# Multimodal interference-based imaging of nanoscale structure and macromolecular motion uncovers UV induced cellular paroxysm

Scott Gladstein[1], Luay M. Almassalha[1], Lusik Cherkezyan[1], John E. Chandler[1], Adam Eshein[1], Aya Eid[1], Di Zhang[1], Wenli Wu[1], Greta M. Bauer[1], Andrew D. Stephens[2], Simona Morochnik[1], Hariharan Subramanian[1,3], John F. Marko[2,4,5], Guillermo A. Ameer[1,3,5], Igal Szleifer[1,3,5] & Vadim Backman[1,3,5]

Understanding the relationship between intracellular motion and macromolecular structure remains a challenge in biology. Macromolecular structures are assembled from numerous molecules, some of which cannot be labeled. Most techniques to study motion require potentially cytotoxic dyes or transfection, which can alter cellular behavior and are susceptible to photobleaching. Here we present a multimodal label-free imaging platform for measuring intracellular structure and macromolecular dynamics in living cells with a sensitivity to macromolecular structure as small as 20 nm and millisecond temporal resolution. We develop and validate a theory for temporal measurements of light interference. In vitro, we study how higher-order chromatin structure and dynamics change during cell differentiation and ultraviolet (UV) light irradiation. Finally, we discover cellular paroxysms, a near-instantaneous burst of macromolecular motion that occurs during UV induced cell death. With nanoscale sensitive, millisecond resolved capabilities, this platform could address critical questions about macromolecular behavior in live cells.

[1] Department of Biomedical Engineering, Northwestern University, Evanston, IL 60208, USA. [2] Department of Molecular Biosciences, Northwestern University, Evanston, IL 60208, USA. [3] The Center for Advanced Regenerative Engineering, Northwestern University, Evanston, IL 60208, USA. [4] Department of Physics & Astronomy, Northwestern University, Evanston, IL 60208, USA. [5] The Center for Physical Genomics and Engineering, Northwestern University, Evanston, IL 60208, USA. Correspondence and requests for materials should be addressed to V.B. (email: v-backman@northwestern.edu)

At the level of individual living cells, thousands of unique molecules are constantly moving, interacting, and assembling-working to execute cellular functions and keep the cell alive. Understanding the properties of this complex motion and its interplay with the cellular ultrastructure remains one of the most critical and challenging topics of study in modern biology. While widely explored, the link between nanoscale structure and molecular motion is particularly challenging to study for several reasons: (1) nanoscale macromolecular organization is often composed of hundreds to thousands of distinct molecules, some of which cannot be easily labeled such as lipids, nucleic acids, or carbohydrates, (2) molecular dynamics depends uniquely on the timescales of interest in the context of the surrounding macromolecular nanostructure, and (3) molecular motion and ultrastructure evolve in concert but along distinct timescales, often spanning milliseconds to days.

Most techniques to study molecular motion in eukaryotic cells require the use of exogenous small molecule dyes or transfection-based fluorophore labeling. These techniques, such as single molecule tracking, fluorescence recovery after photobleaching (FRAP)[1,2], photoactivation[3,4], fluorescence correlation spectroscopy (FCS)[5], and Förster resonance energy transfer (FRET)[6] have greatly expanded our understanding of the behavior of molecular motion in live cells. Despite their utility and the insights produced regarding cellular behavior, these methods have limitations. For instance, single molecule tracking, FRET, and FCS provide information on the activity of individual molecules, but cannot probe the motion of complex macromolecular structure that often govern cellular reactions, such as the supra-nucleosomal remodeling that may occur during gene transcription or DNA replication. Likewise, FRAP and photoactivation can yield diffraction-limited information about the general molecular mobility within cellular compartments, but requires the use of high intensity photobleaching which may damage the underlying structure.

Beyond technique specific applications, these methods share common limitations: (1) they can only probe the behavior of an individual or a few molecules concurrently; (2) they require the use of either potentially cytotoxic small molecule dyes or transfection, which often cannot label lipid or carbohydrate assemblies directly; (3) they are susceptible to artifacts due to photobleaching; and (4) they have significant limitations to probe cellular heterogeneity due to the inherent variability of label penetrance, a critical feature of multicellular systems and diseases, including cancer[7–10]. Further, to extend these techniques to study the interplay between local structure and motion requires the use of additional fluorophores, which have similar drawbacks.

To address these issues, techniques have been developed based on quantitative phase imaging (QPI)[11] and dynamic light scattering (DLS)[12] to image intracellular dynamics without the use of labels. Techniques such as phase correlation imaging[13], magnified image spatial spectrum microscopy[14], and dispersion-relation phase spectroscopy[15] extract diffusion coefficients from temporal fluctuations in phase via the dispersion relation. These techniques have led to interesting biological discoveries, such as a universal behavior where intracellular transport is diffusive at small scales and deterministic at large scales as well as differences in molecular motion between senescent and quiescent cells.

Building upon these advancements, we present a label-free interference-based platform (dual-PWS) that captures the temporal behavior and structural organization of macromolecular assemblies in live cells. This platform is an expansion of live cell Partial Wave Spectroscopy (PWS), a quantitative imaging technology that provides label-free measurements of nanoscale structure[16]. PWS obtains this information by taking advantage of an interference phenomenon in the light backscattered from intracellular macromolecular structures. This interference produces spectral variations that depend on the nanoscale organization of these structures. PWS has resulted in many breakthroughs in the study of the higher-order organization of chromatin structure, its relation to the development of cancer[9,17], and its use in cancer diagnostics[18–23] and therapeutics[10]. The same interference phenomenon that enables PWS to probe intracellular structure at length-scales below the diffraction limit without the use of labels can be utilized to retrieve information on macromolecular motion by measuring temporal variations in the interference signal instead of spectral (wavelength) variations. By combining these two techniques, we pair measurements of cellular dynamics with macromolecular structure—creating a dual light interference platform (dual-PWS) to greatly enhance our understanding of the physical state of the cell and our ability to probe cellular behavior at the level of macromolecular assemblies. While dual-PWS is not molecularly specific, it captures the underlying behavior of all macromolecular assemblies and is complementary to the aforementioned molecular specific techniques[24]. In addition to measuring diffusion coefficients (similar to other label-free techniques such as QPI and DLS), dual-PWS extracts additional quantifications of motion such as the fractional moving mass, which provides a measurement of the volume fraction of moving structures and the mass of moving structures. Beyond that, dual-PWS is sensitive to motion occurring on (or confined to) length-scales smaller than the diffraction limit. Finally, as this method relies on analysis of scattered light, its temporal limitations are defined by current optical technology (illumination intensity, detector sensitivity, etc.) and not by the photochemical limits of probe dyes. Consequently, this interference-based platform measures nanoscale macromolecular motion in tandem with spatio-temporal behavior of the macromolecular ultrastructure in dozens of live cells simultaneously without photobleaching artifacts.

Herein, we present the theory for temporal interference and validate this system using experimental measurements of nanosphere phantoms. Applying dual-PWS microscopy in vitro, we explore nanoscale structural and dynamic changes that occur due to crosslinking chemical fixation and the differentiation of stem cells. Then, we probe the spatio-temporal response of macromolecular assemblies to ultraviolet (UV) light. We discover a phenomenon that occurs early in the process of UV induced cell death: a near-instantaneous burst of motion, referred to in this manuscript as cellular paroxysm. These paroxysms are predominantly asynchronous between cells, are uncorrelated with membrane permeabilization, mitochondrial membrane potential, caspase-3/7 activation, and DNA replication, however, are directly correlated with phosphotidylserine (PS) externalization and depolymerization of the actin cytoskeleton. This nanoscale transformation is synchronous across the cell, originating at loci that are microns apart within 35 milliseconds. This cellular paroxysm indicates the existence of an undiscovered phenomenon that may play a role in the earliest stages of a unique form of cell death. Altogether, we demonstrate that dual-PWS measures nanoscale structure and dynamics with millisecond temporal capabilities, allowing for the label-free quantification of macromolecular motion and structure in live cells. These capabilities enable the discovery of phenomena not previously possible by other techniques.

## Results

**Theory: Origin of back-scattered interference.** The dual-PWS system (Fig. 1a) takes advantage of light interference to amplify a weak scattering signal from cellular structures, allowing us to probe the nanoscale structural organization and dynamics without necessitating exogenous labels in live cells[16,25]. The

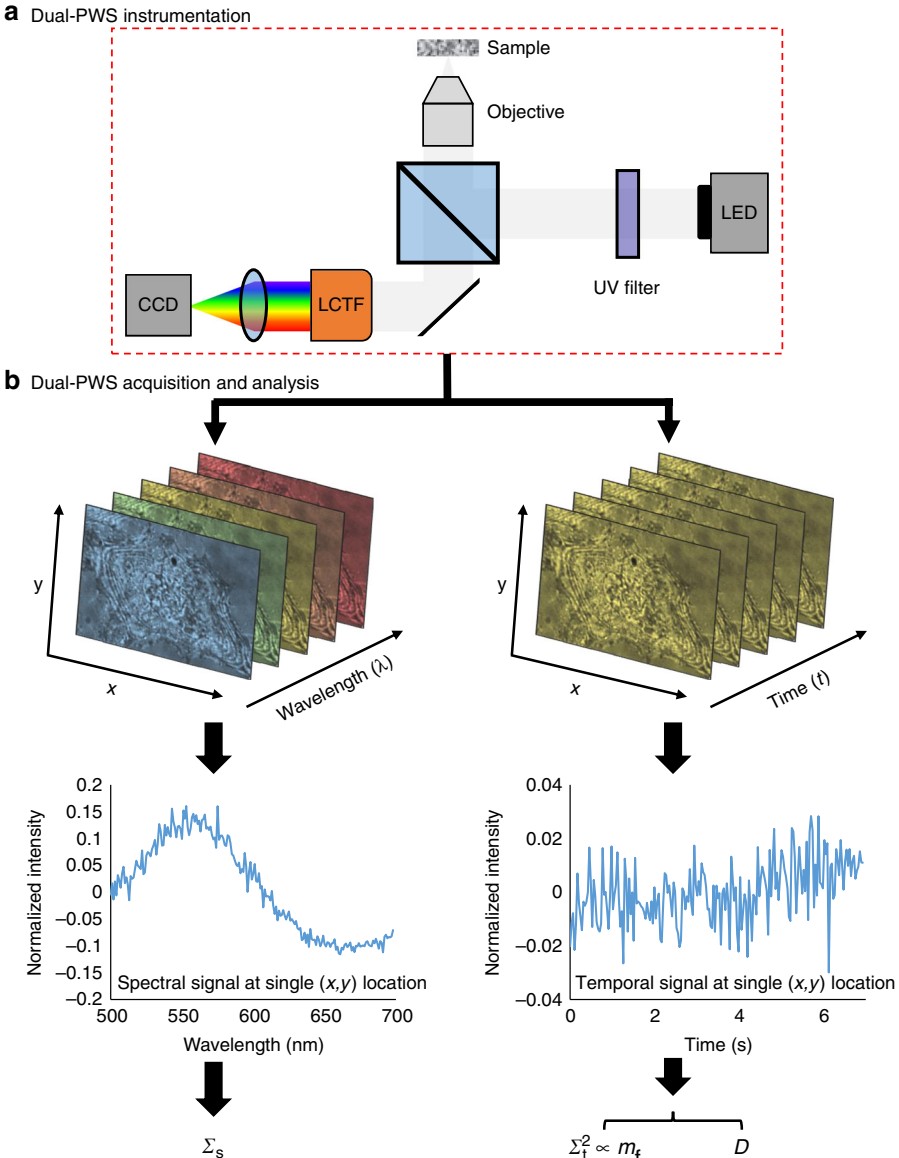

**Fig. 1** Dual-PWS instrumentation, acquisition, and analysis. **a** Schematic of the Dual-PWS instrumentation. Broadband white light from a light-emitting diode (LED) is passed through a filter to remove ultraviolet (UV) components before being focused onto the sample. The backscattered light is collected, spectrally filtered through a liquid crystal tunable filter (LCTF), and imaged with a charge-coupled device (CCD) camera. **b** (left) Structural measurements ($\Sigma_s$) are collected by acquiring multiple backscattered wide-field monochromatic images across a range of wavelengths (500–700 nm) to produce a three-dimensional image cube, $I(\lambda, x, y)$, where $\lambda$ is the wavelength and $(x, y)$ correspond to pixel. $\Sigma_s$ is extracted by calculating the standard deviation of the spectral interference at each pixel $(x, y)$. (right) Dynamics measurements [$\Sigma_t^2$, $m_f$ (fractional moving mass), and $D$ (diffusion coefficient)] are collected by acquiring multiple backscattered wide-field images at a single wavelength (550 nm) over a period of time (acquisition time), to produce a three-dimensional image cube, $I(t, x, y)$, where $t$ is time and $(x, y)$ correspond to pixel positions. $D$ is extracted by calculating the decay rate of the autocorrelation of the temporal interference and $m_f$ is calculated by normalizing the variance of the temporal interference ($\Sigma_t^2$) at each pixel $(x, y)$

interference signal originates from (1) a strong reflectance from the glass-cell interface (petri dish) due to the high refractive index (RI) mismatch, referred to as the reference arm, and (2) scattering from within the cell due to refractive index fluctuations, referred to as internal scattering. This interference signal will vary spectrally, referred to in this paper as spectral interference, depending on the organization of intracellular structures, which generate the internal scattering. Similarly, this interference signal will vary in time, referred to in this paper as temporal interference, depending on the motion of those intracellular structures.

**Theory: Analysis of spectral interference**. PWS microscopy utilizes the spectral interference to measure the structural

organization within cells (Fig. 1b). PWS quantifies the variations in mass density within a sample, $\Sigma_s$, (i.e., the heterogeneity of the macromolecular organization) with a sensitivity to length-scales ranging from 20–200 nm. $\Sigma_s$ is acquired by calculating the standard deviation of the spectral interference. Simulations, further biological explanations, and details on measurement and theory of $\Sigma_s$ can be found in previous works[16,25–29].

**Theory: Analysis of temporal interference**. Several properties of macromolecular motion are simultaneously obtained by capturing the backscattered temporal interference spectrum, of which the two most prominent are (1) the fractional moving mass and (2) the correlation time/diffusion coefficient (Fig. 1b).

The fractional moving mass, $m_f$, is the product of the mass of the typical moving macromolecular cluster, $m_c$, and the volume fraction of mobile mass, $\Phi$, within the sample. This property is acquired by measuring and normalizing the variance of the temporal interference, $\Sigma_t^2$.

$$\Sigma_t^2 = \frac{1}{\Delta T} \int_{\Delta T} \delta I(k,t)^2 dt \propto m_f = m_c \Phi \qquad (1)$$

where $\delta I(k, t)$ is the time-varying part of the intensity reflectance at wavenumber, $k$, and $\Delta T$ is the acquisition time. In order to better understand these parameters, examples of the effect of different $m_c$ and $\Phi$ on fractional moving mass ($m_f$) are explored. In a healthy cell, chromatin moves in a highly dynamic manner, continuously altering its structure and function (large $\Phi$, large $m_f$). However, the addition of chemically crosslinking paraformaldehyde would decrease the fractional moving mass by decreasing the volume fraction of moving chromatin without changing the density or length-scales (same $m_c$, small $\Phi$, small $m_f$) (see Cellular Fixation section for experimental data). Next, a comparison of the effect of molecular length-scales on the fractional moving mass can be made by considering the motion of actin filament bundles in comparison to an equivalent amount of actin monomers. Even though the volume fraction and total mass is equivalent in both samples, each moving structure in the filament sample is larger, and therefore produces a greater fractional moving mass (large $m_c$, large $m_f$) compared to the sample of monomers (small $m_c$, same $\Phi$, small $m_f$). It should be noted that dual-PWS does not specifically measure actin filaments, individual chromatin loops, etc.; instead, it measures the physical and dynamic properties of all macromolecular assemblies at each location inside the cell within the sensitivity range.

In contrast to the fractional moving mass, $m_f$ (quantified through the variance of the temporal interference), the auto-correlation of temporal interference describes how correlated the motion is at different timescales. Physical properties, such as the diffusion coefficient, $D$, can be extracted from the shape of the temporal autocorrelation function. While not all motion within the cell is specifically diffusive, $D$ provides a convenient parameter for measuring the rate of the macromolecular motion. $D$ is acquired by calculating the decay rate of the temporal autocorrelation function as follows

$$B_{\delta I}(\tau, k) = \int_{-\infty}^{+\infty} \delta I(t, k) \delta I(t - \tau, k) dt \qquad (2)$$

$$\frac{B_{\delta I}(\tau, k)}{B_{\delta I}(0, k)} = e^{-\frac{\tau}{t_c}} = e^{-4k^2 D \tau} \qquad (3)$$

$$D = \frac{1}{4k^2 t_c} \qquad (4)$$

where $B_{\delta I}(k, t)$ is the autocorrelation of $\delta I$, $\tau$ is the time lag of the autocorrelation function, and $t_c$ is the correlation time. Future modifications to the dual-PWS instrumentation may enable decoupling of diffusive and deterministic transport by utilizing hyperspectral imaging hardware[15]. A more detailed derivation of the theory behind $m_f$ and $D$ can be found in the Supplementary Note 1.

The system's sensitivity range to molecular motion will depend on the exposure time of each frame for the fastest processes and the length of acquisition for the slowest processes. For a 32 ms exposure and 6.4 s acquisition time (parameters used in phantom experiments), our system is sensitive to processes with diffusion coefficient, $0.065\,\mu M^2 s^{-1} > D > 3.7 * 10^{-5}\,\mu M^2 s^{-1}$ (see Supplementary Note 2). This results in a sensitivity to a broad array of

biological processes within the nucleus, including the diffusion of genomic loci ($\sim 10^{-4}\,\mu M^2 s^{-1}$)[30], transcription factors bound to DNA ($\sim 0.046\,\mu M^2 s^{-1}$)[31], and mRNA ($\sim 0.05\,\mu M^2 s^{-1}$)[32]. Theoretically, the fractional moving mass for isochronic motion is independent of the speed of the moving material, while in practice the measurements will be most sensitive to processes in the middle of the sensitivity range. The system will detect all non-isochronic motion regardless of the speed, but they may not be fully quantifiable as the full signal is not resolved. It should be noted that there are no physical limitations to the speed of processes to which dual-PWS is sensitive. The lower bound of the sensitivity range can be easily adjusted by acquiring more or fewer frames in each acquisition. The sensitivity to faster processes can be improved with currently available hardware modifications to increase the SNR and lower the exposure time (see Supplementary Note 3 and Supplementary Movie 1).

**Nanosphere phantom validation**. To validate the nanoscale sensitivity of the dual-PWS system to molecular motion, a phantom was constructed by suspending carboxyl functionalized polystyrene nanospheres (Phosphorex Inc., RI = 1.60) in a glycerol-water solution (90% glycerol RI = 1.46 or 70% glycerol RI = 1.44). The glass bottoms of the petri dishes were replaced with sapphire (RI = 1.77) to maintain a similar refractive index mismatch for our reference arm (glass-media vs. sapphire-glycerol) in comparison to live cell experiments. Spheres of different sizes and volume fractions were tested to validate physical properties that underlie changes in the fractional moving mass and diffusion in cells (Fig. 2). For fractional moving mass, spheres with radii of 100 nm, 50 nm, 37.5 nm, and 25 nm were used to vary $m_c$, the mass of the typical moving macromolecular cluster, while sphere solutions of different volume fractions (0.1% spheres in 90% glycerol vs. 0.3% spheres in 70% glycerol) were compared to represent changes in $\Phi$. For the most accurate system validation, experimental measurements are compared to the analytical expression of $\Sigma_t$ (from eq. 6, where $\tau = 0$), which removes some of the assumptions used in deriving $m_f$. Additional corrections to account for exposure time are applied (see Supplementary Note 2). As expected, $\Sigma_t$ is sensitive to changes in both $m_c$ and $\Phi$, closely matching theory [$\Delta\Sigma_t = 21.5 \pm 3.5\%$ (SEM), $n = 5$, averaged percent error between experiment and theory, $R^2 = 0.78$] (Fig. 2b). For validation of diffusion coefficients, $D$, experimental measurements of the 0.1% spheres at different sizes are compared to theoretically predicted values using the Stokes-Einstein equation; the measured and theoretical values closely matched for 0.1% solutions [$\Delta D = 13.4 \pm 4.2\%$ (SEM), $n = 5$, averaged percent error between experiment and theory, and $R^2 = 0.993$] (Fig. 2d). As expected, diffusion measurements of the 0.3% solutions are not accurate as the decreased viscosity and exposure time limits result in sphere speeds outside our sensitivity range (Supplementary Figure 3). This loss in sensitivity is compensated for in the $\Sigma_t$ comparison using exposure time correction, but the same cannot be done for measurements of diffusion. Although these spheres are not visible using a traditional wide-field microscope due to the diffraction limit, the interference with our reference amplifies the scattering signal, allowing visualization of these nanospheres and measurement of their dynamic properties using dual-PWS (Supplementary Figure 17).

**Cellular fixation**. To confirm the label-free capacity to obtain information on both cellular dynamics and structure in live cells, we measured the effects of cellular fixation in HeLa cells (Fig. 3a). Under normal conditions, cells demonstrate considerable macromolecular motion throughout the nucleus and the cytoplasm. While living cells exhibit a large fractional moving mass, motion

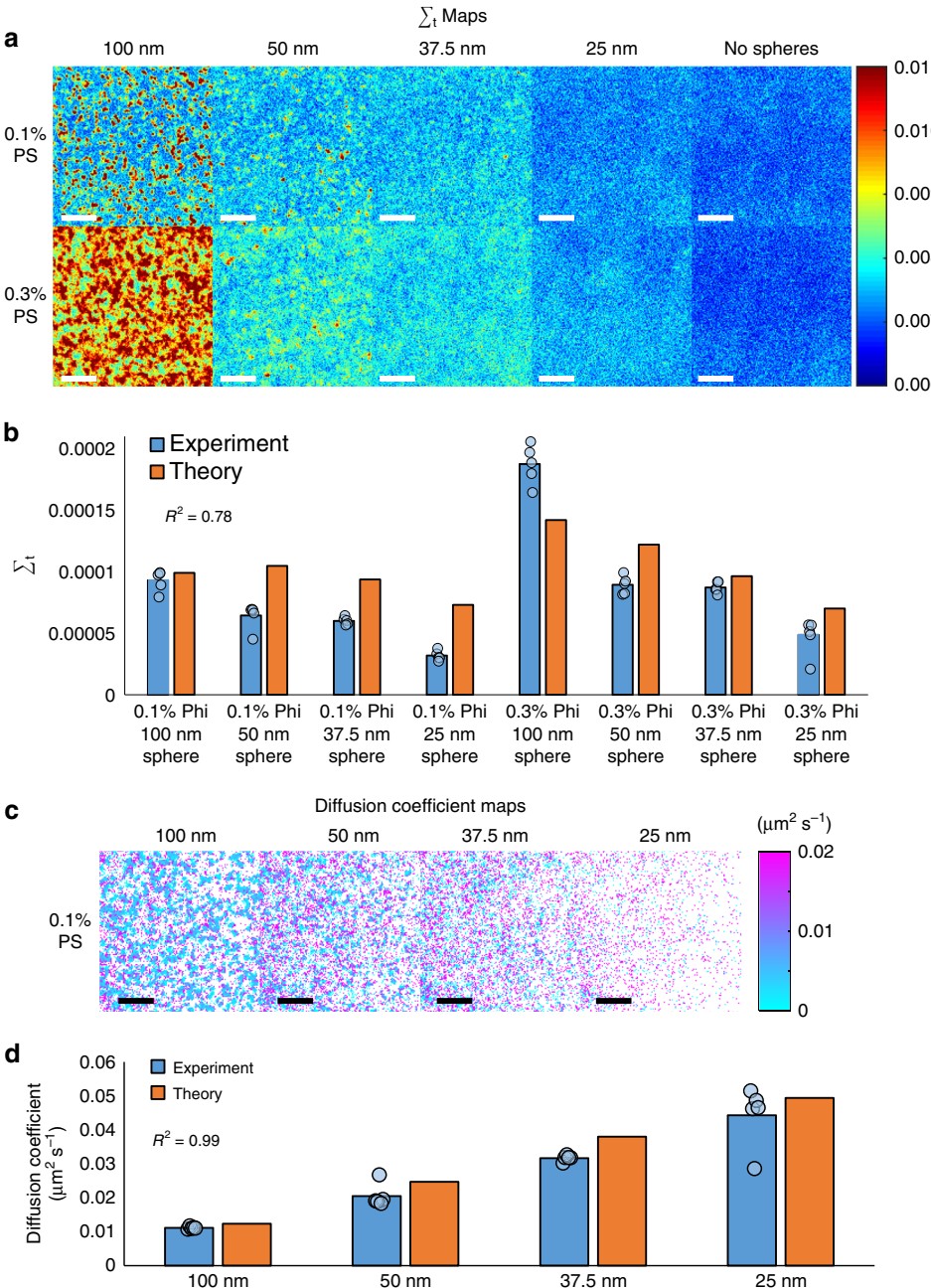

**Fig. 2** Nanosphere phantoms. **a** Representative $\Sigma_t$ maps of polystyrene (PS) nanosphere phantoms created with a variety of nanosphere sizes (25 nm, 37.5 nm, 50 nm, and 100 nm) and concentrations (0.1% and 0.3% volume fractions). **b** Bar graph with individual data points comparing $\Sigma_t$ for the phantoms represented in **a** with analytical theory. The PWS measurements of these phantoms match well with theory ($R^2 = 0.78$) and show that $\Sigma_t$ is sensitive to the volume fraction of spheres ($\Phi$) and the mass of the moving spheres ($m_c$). **c** Representative $m_f$ maps of 0.1% volume fraction polystyrene nanosphere phantoms created with a variety of nanosphere sizes (25 nm, 37.5 nm, 50 nm, and 100 nm) **d** Bar graph with individual data points comparing experimentally measured $D$ with theoretical values calculated using the Stokes-Einstein equation for 0.1% sphere phantoms with sizes 25 nm, 37.5 nm, 50 nm, and 100 nm. The PWS measured $D$ values closely match the predicted values for a correlation of $R^2 = 0.99$. Scale bar is 8 µm

disappears upon crosslinking chemical fixation [10 min, 3.6% paraformaldehyde solution; nucleus, $\Delta m_f = -97.9 \pm 0.8\%$ (SEM), $n = 7$, $p$-value = 0.0006, paired Student's $t$-test; cytoplasm, $\Delta m_f = -95.3 \pm 1.7\%$ (SEM), $n = 7$, $p$-value = 0.0018, paired Student's $t$-test]. The decrease in $m_f$ is likely due to a decrease in $\Phi$ as hypothesized in the Theory: Analysis of Temporal Interference section. Within this experiment, chemical fixation induced alterations in cytoplasm structure [$\Delta\Sigma_s = 33.5 \pm 2.7\%$ (SEM), $n = 7$, $p$-value = 0.0003, paired Student's $t$-test], but preserved the average structural heterogeneity within the nucleus [$\Delta\Sigma_s = -2.3 \pm 3.5\%$ (SEM), $n = 7$, p-value = 0.55, paired Student's $t$-test]

(Fig. 3b). A more thorough examination of the effects of fixation on nanoscale structural heterogeneity has been reported in previous publications[33,34].

**Stem cell differentiation.** The process of cellular differentiation is among the most widely studied in molecular biology and has been demonstrated to result in changes in the organization of chromatin in fixed cells as well as the motility of transcription factors[35,36]. Particularly, Dixon et al. delineated biases in allelic gene expression in embryonic stem cells versus their differentiated progeny; however, the mechanism and rationale for

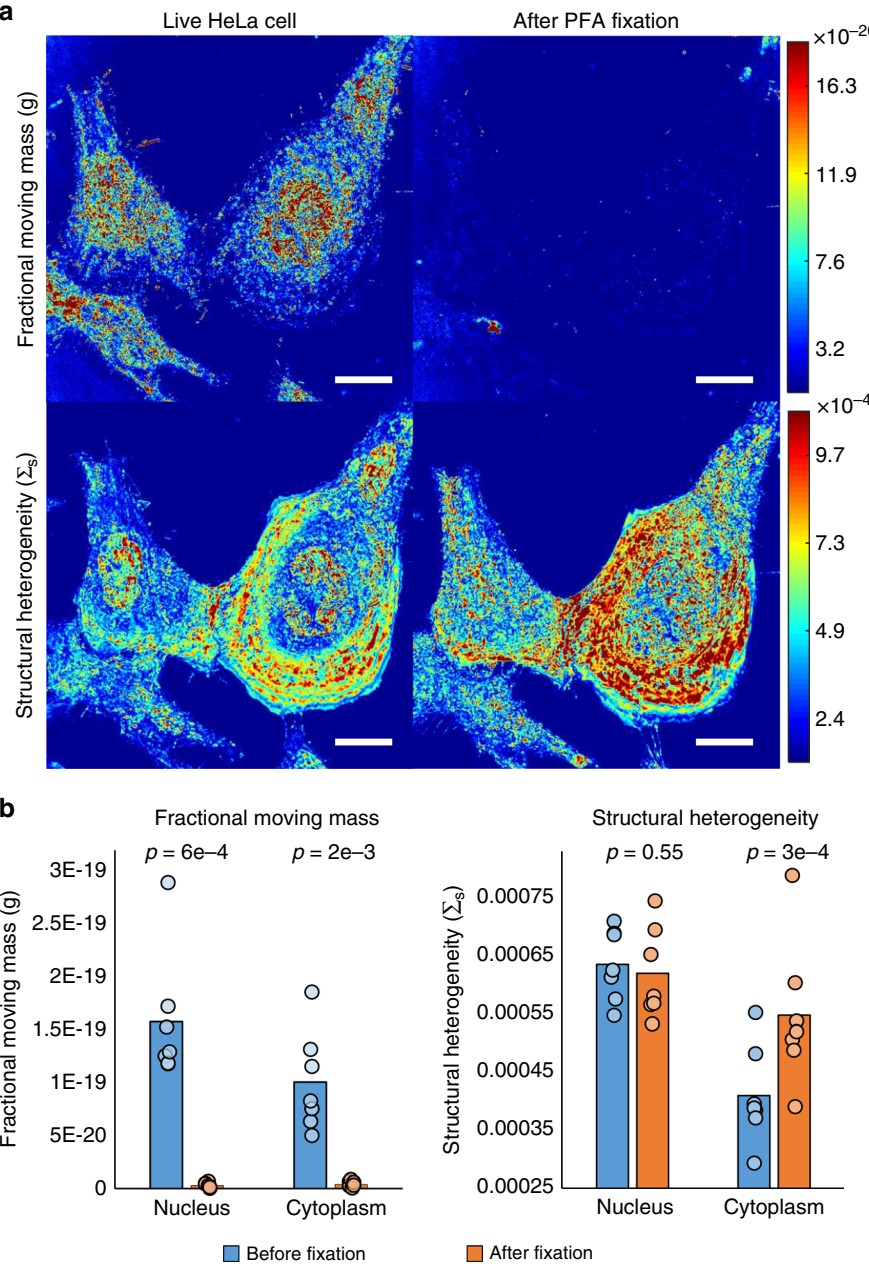

**Fig. 3** Cellular fixation. **a** Representative $m_f$ (top) and $\Sigma_s$ (bottom) maps of HeLa cells before and after crosslinking fixation using paraformaldehyde (PFA). **b** Bar graphs with individual data points quantifying the mean $m_f$ and $\Sigma_s$ for the nucleus and cytoplasm separately. Scale bar is 20 μm. Reported p-values are from two-tailed, paired Student's $t$-tests

higher-order chromatin influence on stem cell differentiation is still a subject of study[36]. Human mesenchymal stem cells (hMSCs) differentiate into different specialized cell types—including adipocytes, chondrocytes, and osteoblasts—due to the integration of a host of chemical and physical signals. Using fluorescence correlation spectroscopy, previous studies have observed slower diffusion of individual transcription factors in stem cells in lineages where their functions are essential[35], but no one has studied large scale changes in motion due to differentiation.

We hypothesized that in hMSCs there would be increased chromatin heterogeneity and higher mobility due to the unique abilities of hMSCs to access genes for multiple phenotypes, given their high differentiation potential. Conversely, we hypothesized that differentiated cells (e.g. osteoblasts) would have lower

heterogeneity and mobility as a result of their committed lineage in which reversion to a stem cell state is no longer likely. Consistent with previous studies, we observe that the hMSC and osteoblast populations have very different states of chromatin folding and nuclear dynamics (Fig. 4a) as well as significant heterogeneity within each population (Supplementary Figure 18). Specifically, hMSCs were observed to have a more heterogeneous chromatin structure [$\Delta\Sigma_s = 25.0 \pm 2.2\%$ (SEM), $p$-value = 9.6e-24, heteroscedastic Student's $t$-test], a higher fractional moving mass [$\Delta m_f = 39.3 \pm 6.0\%$ (SEM), $p$-value = 2.1e-10, heteroscedastic Student's $t$-test], and slower macromolecular motion [$\Delta D = 26.7 \pm 3.4\%$ (SEM), $p$-value = 3.87e-11, heteroscedastic Student's $t$-test] when compared to differentiated osteoblasts ($n = 166$ hMSC and $n = 102$ osteoblasts) (Fig. 4b). These findings support our hypothesis that committed lineages such as osteoblasts lack

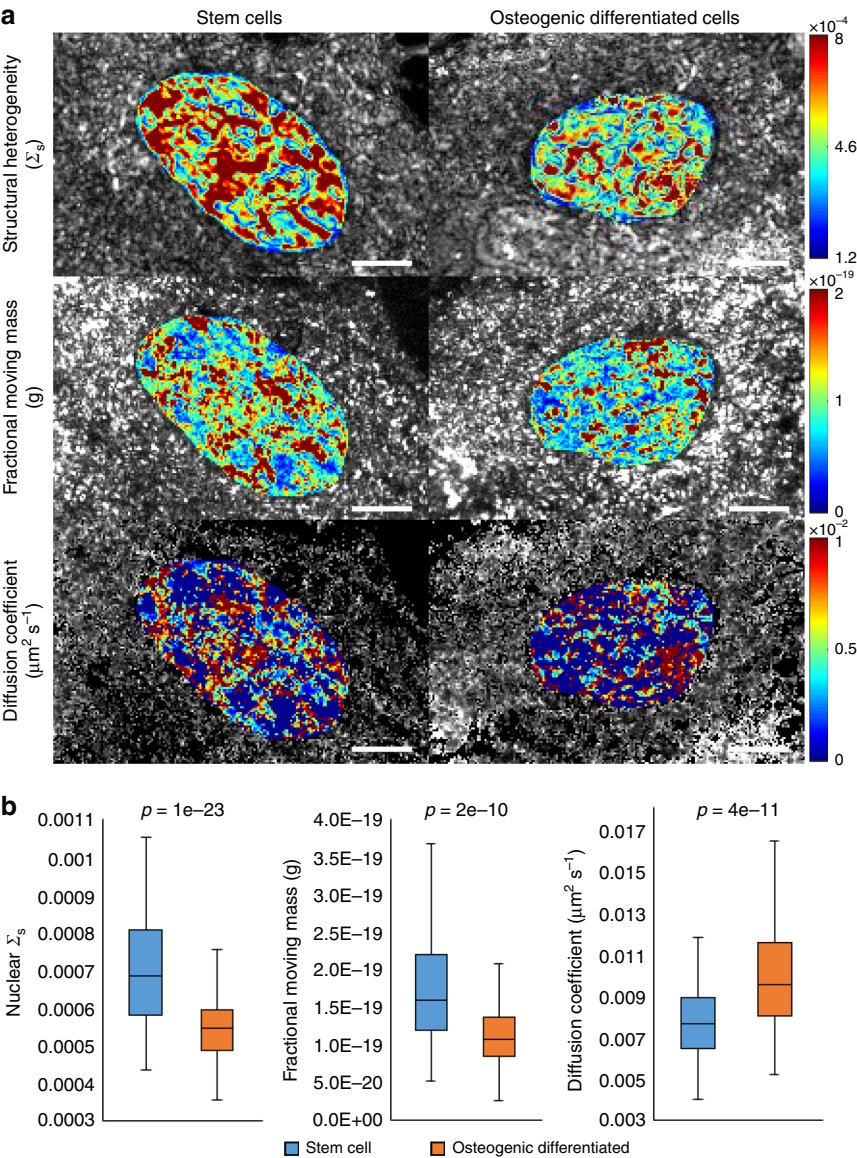

**Fig. 4** Stem cell differentiation. **a** Representative $\Sigma_s$ (top), $m_f$ (middle), and $D$ (bottom) maps of human mesenchymal stem cells (left) and osteoblasts (right) with colored region used to show nuclear segmentation. **b** Box plots showing the distribution of $\Sigma_s$, $m_f$, and $D$ for the stem cells and osteoblasts. We observe different states of chromatin folding and nuclear dynamics comparing hMSCs to osteoblasts. Scale bar is 8 μm. The box within the plot represents the first quartile, median (center line), and third quartile; the whiskers extend to the minimum and maximum values excluding any outliers (1.5x the interquartile range above the third quartile or below the first quartile; outliers not shown). Reported $p$-values are from two-tailed, heteroscedastic Student's $t$-tests

the heterogeneity and molecular motion that potentiate stem cells as well as expand upon previous studies showing that the changes in individual transcription factor diffusion extends to large scale changes in nuclear diffusion. The biological implications of our findings are (1) an improved understanding of the relationship between nanoscale cellular organization and intracellular motion in stem cells may elucidate mechanisms and improve treatment strategies in regenerative medicine, and (2) the differentiation status of stem cells could potentially be determined and/or quantified in live cells in real time without the use of labels to assess stem cell population purity, spontaneous differentiation, and partial vs full differentiation.

To assess the potential of dual-PWS as a non-invasive real-time method to measure differentiation status, we have developed a machine learning model on this dataset to classify cells as hMSCs or osteoblasts based solely on physical properties of the nucleus.

This model was developed using Scikit-learn Logistic Regression classifier[37] and validated using a ten-fold cross validation scheme. This preliminary model has a classification ROC AUC of 88.9 ± 4.1% (SEM) and an accuracy of 80.8 ± 2.6% (SEM). It is likely that this model could achieve higher accuracy by increasing the size of the dataset and developing additional features based on the spatial distribution of these markers and analysis of temporal interference frequencies. Integrating measurements of nanoscale structure and macromolecular motion has provided an enhanced understand of the physical state of the cell enabling applications such as classification of stem cell differentiation status.

**Ultraviolet irradiation.** While UV light is most frequently associated with DNA damage[38], it also has wide ranging effects on the cellular membrane[39], mitochondrial behavior[40], and

chromatin structure[16]. To date, not much is known about the real-time effects on nanoscale (20–350 nm) cellular structure and dynamics due to limitations in existing microscopic techniques as UV illumination can distort the optical properties of dyes as well as induce significant structural damage to cells. Previous work with live cell PWS microscopy to detect alterations in higher-order chromatin structure[41] after UV irradiation demonstrated previously unobserved transformation in cellular and chromatin ultrastructure in real-time. In this study, we apply our dual-mode system to monitor both the transformation to cellular ultra-structure and macromolecular dynamics during UV illumination. To perform UV irradiation, the cells were illuminated with the full LED spectra, resulting in UV irradiation of ~0.013 µW µm$^{-2}$ between 350 and 400 nm. For control measurements, a UV filter is placed in the system to block all light below 405 nm. In control cells, the structure and diffusion coefficients are stable throughout the 25 min experiment; the fractional moving mass is mostly stable, but does show a slight decrease [Nucleus: $\Delta m_f = -12.9 \pm 7.0\%$ (SEM), $n = 28$, $p$-value = 0.003; Cytoplasm: $\Delta m_f = -6.4 \pm 2.7\%$ (SEM), $n = 28$, $p$-value = 0.001; paired Student's $t$-test between time point $t = 0$ and $t = 25$ min] (Figure 5a, b). While the structure on average remains stable, localized structure con-tinuously evolves consistent with the large fractional moving mass detected with the temporal measurements (Supplementary Movie 2). At the level of the nucleus, this indicates that even at very short timescales (milliseconds), supra-nucleosomal organi-zation is continuously evolving.

Interestingly, continuous UV irradiation induces a steady decrease in the fractional moving mass ($\downarrow m_f$) within the first ten minutes of irradiation, ultimately stabilizing in a state of low dynamics for the following fifteen minutes. Structurally, this attenuation in the fractional moving mass is first accompanied by a sharp homogenization in chromatin organization ($\downarrow\Sigma_s$) for the first three minutes, followed by a variable duration plateau spanning from three to eight minutes that precedes a further homogenization of chromatin structure ($\downarrow\Sigma_s$) for the remainder of the experiment (Fig. 5a). The diffusion coefficient shows an increase in the rate of molecular motion for the first 10 min of UV irradiation; the diffusion coefficient was not analyzed for the entirety of the experiment due to low SNR at later time points (see Supplementary Note 4 and Supplementary Figure 5). The structural response in the cytoplasm is distinct from the nuclear response. Within the cytoplasm, there are no structural alterations for the first ~4.5 min. Then, there is a rapid homogenization of structure ($\downarrow\Sigma_s$) for the next ~4.5 min, before the signal stabilizes for the remainder of the experiment (Fig. 5b). Notably, once the cells reach their low dynamic state, the localized structural alterations stop and the spatial distribution of chromatin organization is fixed for the rest of the experiment (Supplementary Movie 2).

Taking advantage of capabilities of live cell PWS microscopy to analyze structure and dynamics in the same cell without photobleaching, we examine the temporal responses of individual cells due to UV irradiation (Figure 5c, d). In particular, a near-instantaneous, large scale burst of motion (transient increase in $m_f$ across the whole cell), termed a cellular paroxysm, occurs between 10 and 20 min that is predominantly asynchronous from cell to cell (Fig. 6a; Supplementary Movie 2 between 00:04 and 00:06). Critically, this analysis is possible due to the dual-PWS system's capacity to perform single cell tracking as we observe a unique stochastic phenomenon that is not apparent in the aggregate behavior of the population. From these measurements, it is clear that these bursts of activity generally occur within a single acquisition (<7 s).

By examining the temporal interference (intensity as a function of time at each pixel) of each cellular event, we observe that the initiation of these bursts of motion can happen in as little as a

single 35 ms frame and can occur independently at distances over 30 µm apart. To quantitatively analyze this phenomenon, we enhanced the temporal resolution to a single frame using spectral feature detection (Figure 6b) and localized the motion burst based on their temporal initiation (Fig. 6d). Figure 6c shows a histogram of the timing of the event for each pixel within a single cell obtained using spectral feature detection (MATLB findchangepts function); surprisingly, we found that within this particular cell, following initiation, 1% of the events occur with the first 35 ms, 19% within 70 ms, and over 50% within 140 ms. An interesting feature of this spatio-temporal organization is that typically these events do not originate in a single location and spread out in a wave throughout the cell as one might assume (Fig. 6d and Supplementary Movie 3). The initial motion occurs simultaneously within 35 ms on opposite ends of the cell and extends into the cell nucleus; to accomplish this, a molecular regulator synchronizing these events would need to diffuse at a rate $D > 3500 \frac{\mu M^2}{sec}$. Consequently, this motion is orders of magnitude faster than protein diffusion ($\sim 20 \frac{\mu M^2}{sec}$) in the eukaryotic cytoplasm. Additional characterization of the behavior of the cellular paroxysm is included in Supplementary Note 5.

As this is a newly observed phenomenon, we explored molecular transformations that accompanied these cellular paroxysms. Owing to the stochastic nature of these events, UV irradiation of cells was performed for an intermediate duration in order to differentially induce the response in some, but not all, cells within the field of view. Using this approach, differential molecular analysis could be performed to examine the differences between cells that underwent the cellular paroxysm (referred to as CP cells) and cells that did not (referred to as NCP cells) under comparable UV illumination. As these bursts of activity occurred throughout the entire cell, we hypothesized that these events were occurring at the level of either the cytoskeleton (actin disruption) or the plasma membrane (pore formation or phosphatidylserine externalization). Alexa Fluor 488 Phalloidin, which binds to F-actin, enables imaging of the cytoskeletal structure. A decrease in Phalloidin fluorescence intensity was observed for cells irradiated with UV with a larger decrease in CP cells [$-81.3 \pm 0.7\%$ (SEM); $n = 18$ (CP) and 22 (controls), $p$-value = 2.8e-17, heteroscedastic Student's $t$-test] compared to NCP cells [$-71.0 \pm 2.7\%$ (SEM); $n = 20$ (NCP) vs 22 (control), $p$-value = 1.4e-19, heteroscedastic Student's t-test] (Fig. 7a). Since Phalloidin binds to and stabilizes F-actin filaments, the decrease in fluorescence intensity indicates F-actin depolymer-ization throughout the cell. Cytoskeletal reorganization has been linked to phosphatidylserine (PS) externalization (a process generally associated with apoptosis) suggesting a potential link between cellular paroxysm and plasma membrane integrity[42,43].

In healthy intact cells, PS is located on the interior of the plasma membrane. Annexin V was used to detect PS externaliza-tion since it cannot pass through an intact cell membrane, therefore, it will only bind to PS which has translocated to the exterior of the membrane (PS externalization). PS externalization is one of the earliest detectable processes in cells that have started to undergo apoptosis; although, there are some non-apoptotic causes of PS externalization such as engagement of immunor-eceptors, ultrashort electric pulses, etc.[43,44]. One of the most interesting biological results was a strong correlation between cellular paroxysm and Annexin V staining. 91% of the cells followed the trend where CP cells were Annexin-V-positive and NCP cells were Annexin-V-negative (26 cells CP and Annexin-V-positive, 25 cells NCP and Annexin-V-negative, 5 cells NCP and Annexin-V-positive, 0 cells CP and Annexin-V-negative) (Fig. 7b). Finally, propidium iodide (PI) was used to detect poration of the plasma membrane. PI is an intercalating agent and fluorescent molecule that cannot enter cells with intact

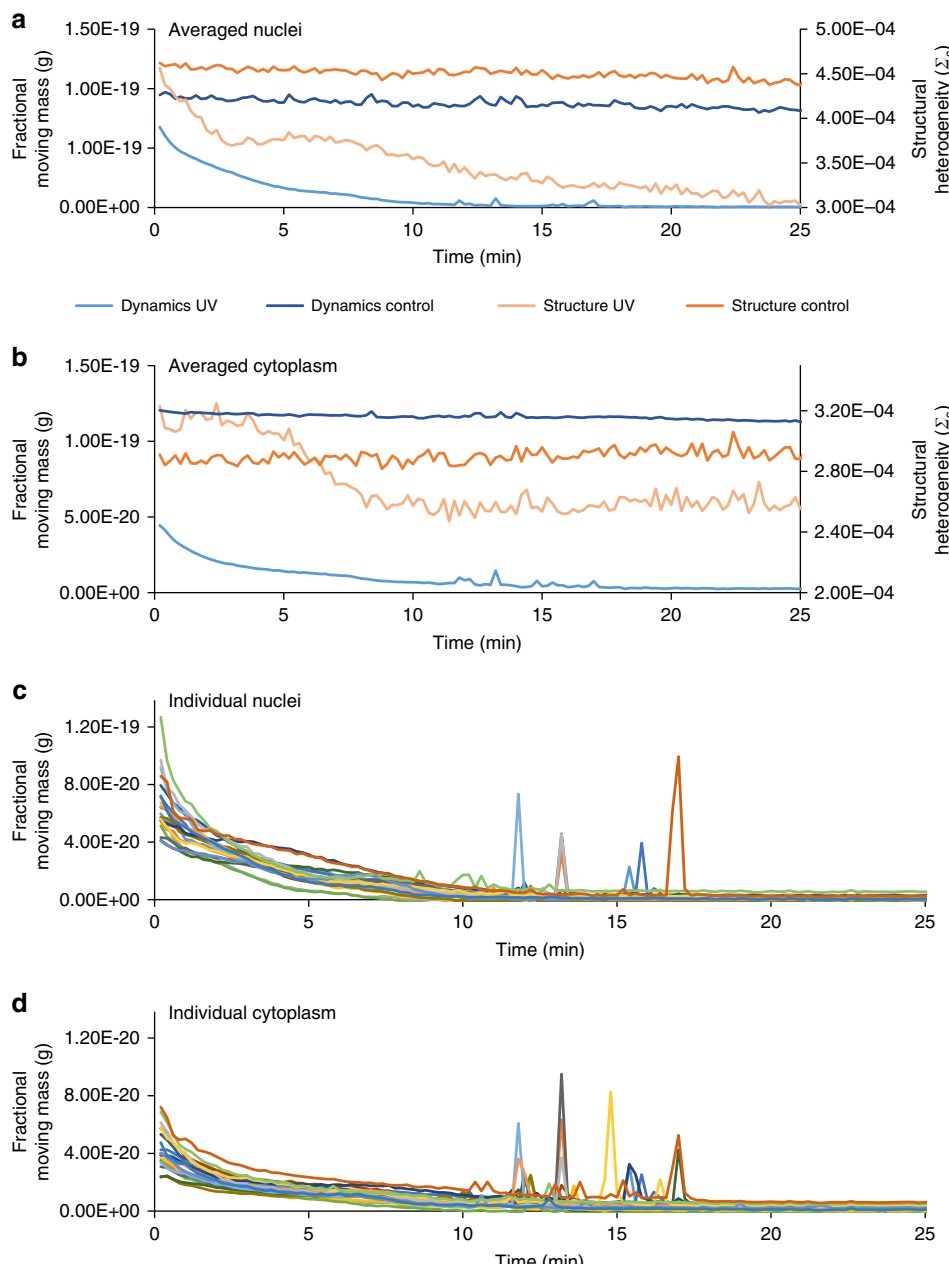

**Fig. 5** Ultraviolet irradiation quantification. Line graphs showing the averaged $\Sigma_s$ and $m_f$ in response to ultraviolet (UV) ($n = 20$) and non-UV ($n = 28$) irradiation within the nucleus (**a**) and the cytoplasm (**b**). Line graphs showing individual cell response to UV irradiation within the nucleus (**c**) and the cytoplasm (**d**). By examining individual cells, a near-instantaneous, stochastic burst of motion, termed a cellular paroxysm, occurs between 10 and 20 min that is asynchronous from cell to cell

plasma membrane. All cells exhibited positive PI staining at the end of the experiment, indicating that although as expected UV caused poration of the cell membrane, poration was not associated uniquely with cellular paroxysm (Fig. 8a). In summary, this burst of motion is connected to a disruption of the cytoskeletal and cell membrane structures, and may play a role in the early stages of cell death.

UV irradiation is known to stall DNA replication. This halt of DNA replication seems to coincide with the arrest of nuclear motion detected with dual-PWS microscopy. To confirm these results, after UV irradiation (and arrest of nuclear motion), cells were stained with Bromodeoxyuridine (BrdU), a synthetic nucleoside that incorporates into freshly synthesized DNA in replicating cells. In control cells, we observed incorporation of BrdU into the nucleus, indicating active DNA replication

(Fig. 8b). In contrast, there was minimal incorporation of BrdU into the nuclei of UV irradiated cells. Additionally, we saw no difference in BrdU incorporation between CP and NCP cells. These results confirm that the UV irradiation in these experiments is stalling DNA replication, but this was independent of the cellular paroxysm occurring within the nucleus. Additional experiments exploring the relationship between UV irradiation, cellular paroxysms, and features of cell death (mitochondrial membrane potential, caspase-3/7, and morphology) are included in Supplementary Note 6.

## Discussion

In this work, we present an imaging platform that integrates the nanoscale structural measurements of PWS microscopy with measurements of nanoscale cellular dynamics. By capturing the

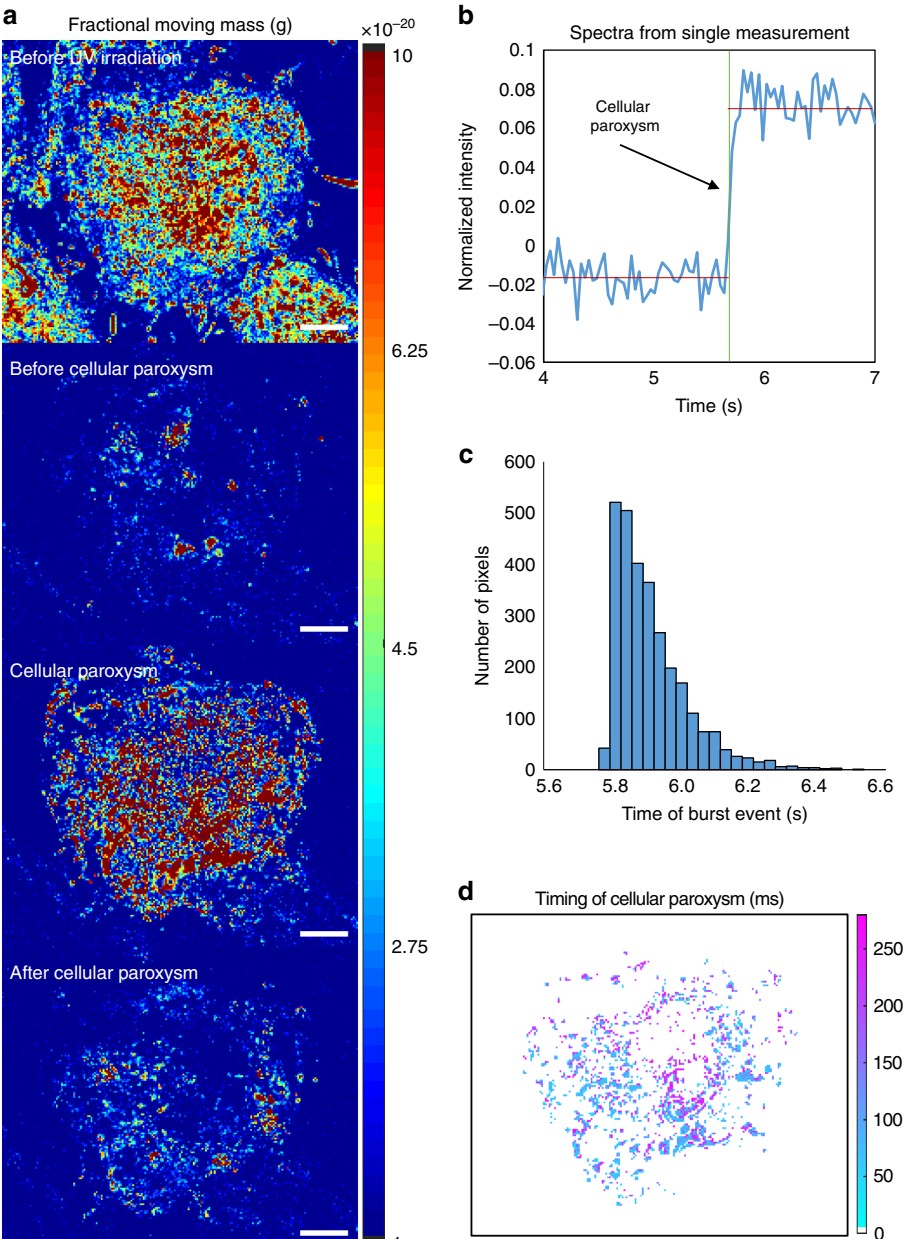

**Fig. 6** Cellular paroxysm temporal analysis. **a** Representative $m_f$ maps of a cell undergoing the cellular paroxysm. **b** Example temporal spectra from a single pixel during a cellular paroxysm. Using spectral feature detection the timing of the event can be determined for each pixel reducing the temporal resolution from the full acquisition time (~7 s) to a single exposure time (35 ms). **c** Example histogram of the timing of the event for each pixel within a single cell. **d** Map displaying the timing of the event for each pixel within a representative cell. The spatial distribution of this timing shows that the initial motion occurs simultaneously within a single 35 ms frame on opposite ends of the cell. Scale bar is 7 μm

temporal interference, this technique quantifies cellular dynamics, such as the fractional moving mass and the diffusion coefficient. First, we validate this instrument comparing measurements of nanosphere phantoms to the analytical theory. As a biological control, we demonstrate that macromolecular motion ceases upon PFA fixation. We explored the structural and dynamic changes that occur in stem cells due to differentiation, demonstrating both increased variations in chromatin folding and molecular motion in the stem cell population. Finally, we studied the effects of UV irradiation and discovered a biological phenomenon, where cells undergo a large rapid burst of motion prior to cell death.

Studying structural and dynamic alterations in stem cells in response to differentiation is an ideal model to explore the processes associated with cells that have undergone a major phenotypic transformation. Our lab has recently used live cell PWS to identify compounds that reduce variations in supra-nucleosomal chromatin folding and thus render cancer cells more susceptible to chemotherapy treatment[10]. The ability to find compounds that modulate the chromatin folding of cells along the stem continuum—that is, to find compounds that increase the rate of differentiation or that trigger de-differentiation—would prove to be crucial in better understanding the stem cell differentiation process and in guiding our search for compounds that could be used to treat patients with neurological disorders.

The results from the UV experiments provide high-temporal resolution, nanoscopic information about cellular motion during the UV irradiation. Interestingly, there are multiple phenomena

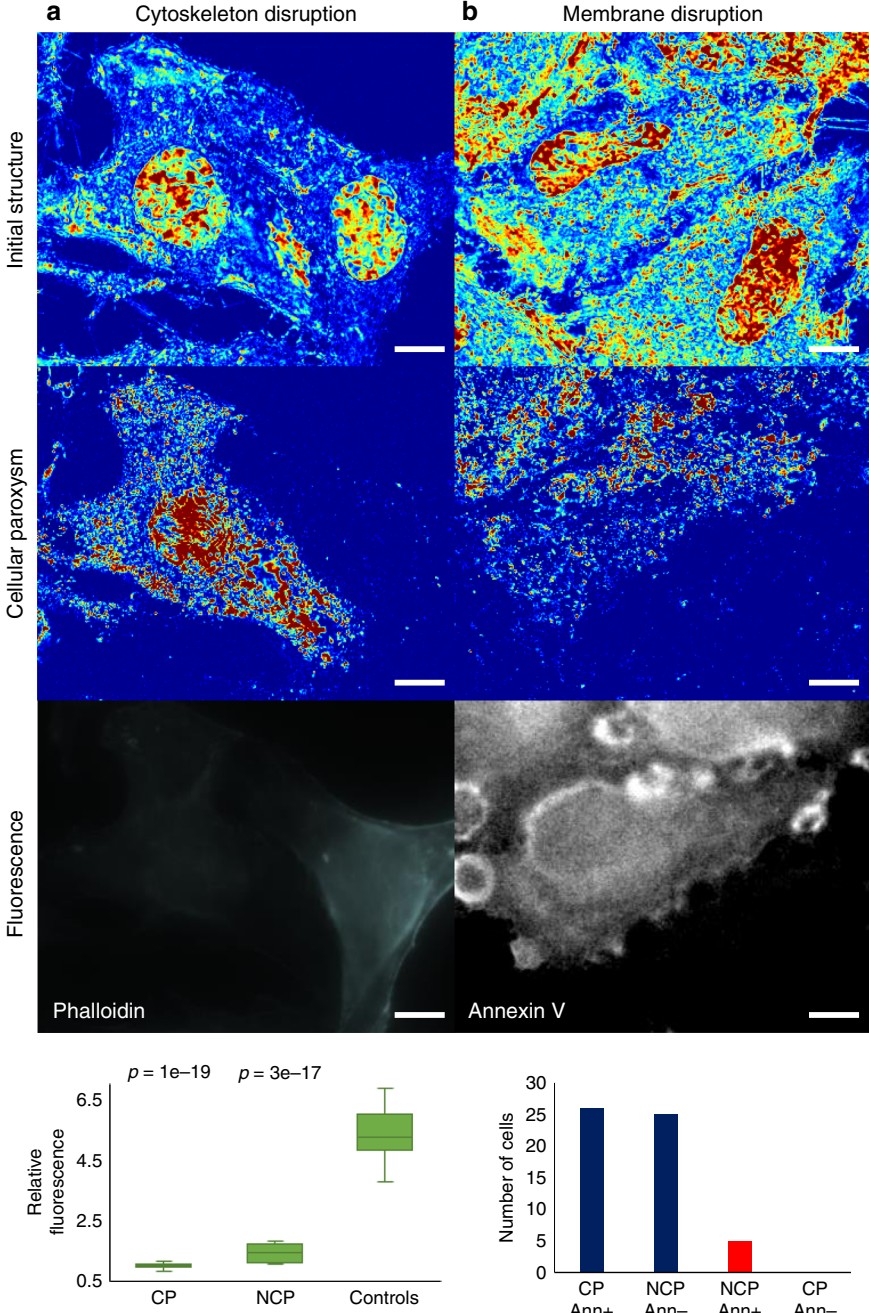

**Fig. 7** Cellular paroxysm biological exploration. Exploration of cytoskeletal (**a**) and membrane (**b**) disruption during cellular paroxysm using Alexa Fluor 488 Phalloidin and Annexin V-FITC, respectively. (row 1) Representative $\Sigma_s$ maps of cells before UV irradiation, scaled from 0.00012–0.00079. (row 2) Multiple $m_f$ maps processed and combined to show which cells experienced the cellular paroxysm event throughout the experiment. (row 3) Representative fluorescence image of Fluor 488 Phalloidin (**a**) and Annexin V-FITC (**b**). (row 4) **a** Box plot showing distribution of the normalized phalloidin intensity and **b** bar graph counting cells that are grouped based on their Annexin V state [positive (Ann+) or negative (Ann−)] and whether they experienced the cellular paroxysm (CP) or not (NCP). The cellular paroxysm is correlated to both cytoskeletal and membrane disruption. The box within the plot represents the first quartile, median (center line), and third quartile; the whiskers extend to the minimum and maximum values excluding any outliers (1.5x the interquartile range above the third quartile or below the first quartile; outliers not shown). Scale bar is 11 μm. Reported $p$-values are from two-tailed, heteroscedastic Student's $t$-tests

observed that are the result of light mediated cellular stress. Initially, we see a homogenization of the chromatin structure, which is likely due to UV induced DNA fragmentation that is paired with an arrest in cytoplasmic and nuclear motion. The observed arrest in motion in the cytoplasm converges with actin depolymerization (Fig. 7a) (consistent with decreased cytoplasmic $\Sigma_s$) and poration of the cell membrane (Fig. 8a). After cells reach

a near static state, a stochastic eruption of motion (cellular paroxysm) forms across the entire cell. While the cellular paroxysm is asynchronous from cell to cell across the field of view, it frequently synchronizes for cells in contact with each other. Temporal analysis of events within individual cells show that these events can occur across the entire cell as far as 30 μm apart in under 35 ms.

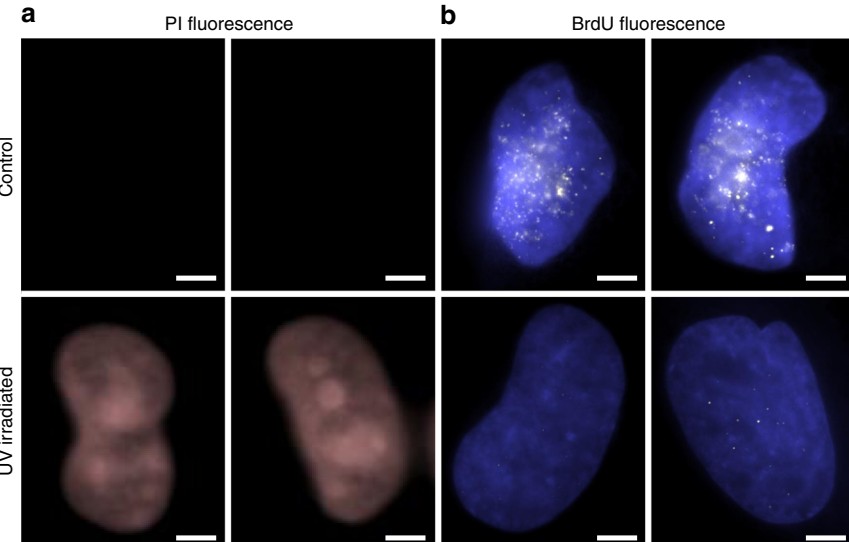

**Fig. 8** Membrane poration and DNA replication. **a** Representative fluorescence images of propidium iodide (PI) for control cells (top) and UV irradiated cells (bottom) show that pores form in the cell membrane in response to UV irradiation. **b** Representative fluorescence images of Bromodeoxyuridine (BrdU) for control cells (top) and UV irradiated cells (bottom) show that UV irradiation is stalling DNA replication. Scale bar is 5 μm

To be synchronous and diffusive in nature, a molecular regulator would need to move at an incredibly high speed $(D > 3500\,\mu M^2 s^{-1})$ to facilitate this process. Molecularly, the cellular paroxysm correlates with PS externalization and actin depolymerization. The correlation with actin depolymerization has two possible explanations. The first explanation is that part of this event is an additional rapid depolymerization of actin. As actin accounts for 5–10% of all protein within the cell, spontaneous alteration of that entire structure would produce a significant movement of mass throughout the entire body of the cell[45]. Alternatively, actin depolymerization is a precursor to this event, and therefore cells, which depolymerize actin faster, will undergo the cellular paroxysm first. Interestingly, the cellular paroxysm also takes place within the nucleus indicating either a direct integration of chromatin motion with actin depolymerization at short timescales or a separate determining mechanism.

In addition to the coupling of cellular paroxysm with actin depolymerization, there is a strong correlation with PS externalization (100% of CP cells are Annexin-V-positive and 83% of NCP cells are Annexin-V-negative). This suggests a synchronous coupling between cellular paroxysm, cytoskeletal organization, and membrane structure at short timescales. With respect to the cells that were uncorrelated (9% of cells), these cells displayed positive Annexin V staining without an accompanying cellular paroxysm. Notably, there were no CP cells that displayed negative Annexin V staining. The likely explanation is that these uncorrelated cells did actually experience cellular paroxysm, but the events were not captured. One reason that we may have missed a few of these events is that there is a short time gap (<20 s) between the completion of our final measurement in these experiments and the UV light being turned off. From these results, PS externalization and the cellular paroxysm occur either simultaneous or potentially PS externalization occurs a few seconds before. Due to limitations in the Annexin V staining and imaging procedure, the precision of this timing cannot be further increased as we have to complete the PWS imaging series before imaging the Annexin V. It is difficult to speculate on the magnitude of fractional moving mass that a PS externalization event would cause as the exact physical mechanism is unknown, but the independent motion of PS would produce a weaker signal than large scale actin depolymerization. These experiments suggest

that both of these molecular transformations are linked to a structural transformation in the cell at short timescales.

While these studies suggest that cellular paroxysm could be an early event in UV mediated cell death, the exact mechanism of cell death is unclear. UV irradiation is inducing cell death as observed through molecular processes such as PS externalization, membrane permeabilization, loss of mitochondrial membrane potential (Supplementary Figure 10), and actin depolymerization, as well as a disruption in cell function as observed through a lack of intracellular motion [at the time of paroxysm and up to 20 h later (see Supplementary Note 7 and Supplementary Figure 10)], macroscopic cell movement within the field of view, and DNA replication (at time of paroxysm). The loss of mitochondrial membrane potential and the actin depolymerization are consistent with apoptotic and necrotic pathways, but some of these features do not follow the traditional pathways (Table 1). Typically, healthy viable cells are characterized by Annexin V-/PI-, cells in early apoptosis are characterized by Annexin V+/PI−, and necrotic or late stage apoptotic cells are characterized by Annexin V+/PI+. In these experiments, after UV irradiation, cells are exhibiting Annexin V−/PI+ before cellular paroxysm and Annexin V+/PI+ after cellular paroxysm. One possibility is that we are observing early necrotic cells in a short time window where the holes in the plasma membrane are large enough for PI to pass through, but not yet large enough for Annexin V. Otherwise, we are potentially observing an alternative form of necrosis. Within traditional apoptosis and most pathways of necrosis, cells exhibit significant morphological changes, such as cell shrinkage, rounding, membrane blebbing, and eventual cellular fragmentation. While macroscopic morphological changes are observed in cells after short dosages of UV irradiation (1 min), no morphological changes are observed within 20 h in cells irradiated for more than 6 min, with or with a cellular paroxysm (Supplementary Figure 12). Additionally, activated caspase-3/7 was not detected in any of the irradiation conditions at 2.5 or 20 h (Supplementary Figure 11). In conclusion, while these cells are undergoing a form of cell death, they are not following traditional apoptotic or necrotic pathways (Table 1).

In summary, we developed a system that enables label-free live cell measurements of nanoscale structure and macromolecular motion with millisecond temporal capabilities, derived theory for

**Table 1 Features of cell death and UV induced cellular paroxysm**

| | PS externalization | Membrane permeability | Actin depolymerization | Mitochondrial membrane potential | Caspase-3/7 activation | Morphological changes (<3 h) | Morphological changes (12–24 h) |
|---|---|---|---|---|---|---|---|
| >6 min UV irradiation + cellular paroxysm | Yes | Yes | High | Low[a] | No[b] | No | No |
| >6 min UV irradiation without cellular paroxysm | No | Yes | Medium | Low[a] | No[b] | No | No |
| 1–3 min UV irradiation without cellular paroxysm | No | Not tested | Not tested | Medium[a] | No[b] | Yes | Yes |
| Early stage apoptosis | Yes | No | Medium | Low | Yes | No | NA |
| Last stage apoptosis | Yes | Yes | High | Low | No | NA | Yes |
| Necrosis | Yes | Yes | High | Low | No | Varied | Varied |
| Healthy cells | No | No | Low | High | No | No | No |

This table summarized different features of the apoptotic and necrotic processes and if these features occur in UV irradiated cells with and without cellular paroxysm
[a]Measured at 2.5 h after UV irradiation
[b]Measured at 2.5 and 20 h after UV irradiation

measurements of temporal interference, and validated the system using experimental phantoms. Using this dual-PWS system, we explored macromolecular structure and motion during stem cell differentiation and the cellular response to UV irradiation. As a high-speed imaging technique, dual-PWS can track individual cells in real time, enabling the identification of asynchronous changes (like the cellular paroxysm), comparisons of the same cells, before, after, and during treatments, and detection of alterations in sub-cell populations.

This technique could provide valuable information about structure-function relationships within the nucleus, especially for the important processes of repair, replication, and transcription. In particular, it can facilitate investigation about the spatio-temporal biological regulators of chromatin, such as: does the cytoskeleton or the nuclear lamina regulate chromatin, and what role does the intranuclear electrostatic environment play? With respect to human diseases, this technique allows investigation into therapeutics that regulate chromatin structure and dynamics in fields such as regenerative medicine, cancer, and infectious diseases. While chromatin is of particular interest as the regulator of transcription, replication, and repair, this technology can be used to study any biological systems, such as cytoskeletal dynamics during the processes of cell division, muscle contraction, or carcinogenesis.

## Methods

**Microscope design and data acquisition**. The dual-PWS system consists of a commercial microscope base (Leica DMIRB) equipped with a broad-spectrum white light LED source (X-Cite 120LED), long pass filter (Semrock BLP01-405R-25), 63x oil immersion objective (Leica HCX PL APO, NA 1.4 or 0.6), spectral filter (CRi VariSpec LCTF), and CCD camera (Hamamatsu Image-EM CCD). With the microscope in epi-illumination mode, light from the LED is passed through a long pass filter to remove UV components before being focused by the objective onto the sample with an illumination NA of 0.55. The backscattered light is collected by the objective, passed through the spectral filter (LCTF) and imaged at the CCD (Fig. 1a). Multiple wide-field monochromatic images are obtained for each acquisition and stored in a three-dimensional image cube as described below.

Structural measurements ($\Sigma_s$) are collected by acquiring multiple backscattered wide-field monochromatic images across a range of wavelengths

(500–700 nm) to produce a three-dimensional image cube, $I(\lambda, x, y)$, where $\lambda$ is the wavelength and $(x, y)$ correspond to pixel positions (Fig. 1b). Each frame within a spectral data cube is acquired with a 35 ms exposure, and the total cube can be obtained in under two seconds depending on the number of wavelengths collected[24]. As described above, $\Sigma_s$ is extracted by calculating the standard deviation of the spectral interference at each pixel $(x, y)$. Similarly, dynamics measurements ($\Sigma_t^2$, $m_f$, and $D$) are collected by acquiring multiple backscattered wide-field images at a single wavelength (550 nm) over a period of time (acquisition time), to produce a three-dimensional image cube, $I(t, x, y)$, where $t$ is time and $(x, y)$ correspond to pixel positions (Fig. 1b). $D$ is extracted by calculating the decay rate of the autocorrelation of the temporal interference and $m_f$ is calculated by normalizing the variance of the temporal interference ($\Sigma_t^2$) at each pixel $(x, y)$; Further details on these calculations are included in Supplementary Note 1. To correct for temporal LED fluctuations (and other system noise), reference dynamics image cubes are collected from an empty field of view within the sample; $\Sigma_t^2$ calculated from the reference cube is subtracted from our sample's $\Sigma_t^2$ to remove noise contributions. For this study, temporal measurements consisted of at least 201 frames acquired with 1.2 ms, 32 ms, or 35 ms exposures each for a total time of ranging between 240 ms to 7 s (Table 2). In theory, the exposure and acquisition time can be adjusted to match the biological phenomenon of interest (see Supplementary Note 2). In the current configuration, the temporal and structural measurements are performed sequentially in <15 s. See Supplementary Note 8 for a discussion on how hardware and imaging parameters will affect SNR. See Table 2 for a list of the experimental parameters used for each experiment.

**General diffusion analysis**. Diffusion coefficients were calculated in the following manner:

(1) Temporal interference data cube is loaded and normalized for LED intensity, exposure time, and dark counts.
(2) The temporal mean is subtracted from each pixel so that the temporal oscillations fluctuate around zero.
(3) The temporal autocorrelation function (ACF) is calculated at each pixel.
(4) ACFs with low SNR are removed. This is determined by pixels where the first point of the ACF is less than $\sqrt{2}$ multiplied by the first point of the ACF of the background (region without cells).
(5) The average ACF of a background sample (no cell or spheres) is subtracted from each individual ACF.
(6) The ACF at each pixel is normalized so that the first point equals one.
(7) Negative values are removed in order to calculate the natural log of the ACF at each pixel.
(8) The natural log of the ACF is averaged across all pixels. The mean ACF can be obtained by inversing the natural log using the exponential function, although this is not necessary to calculated $D$.

## Table 2 Experimental imaging parameter

| Experiment | Illumination NA | Collection NA | Exposure time | Acquisition time | Imaging wavelength |
|---|---|---|---|---|---|
| Nanosphere phantoms | 0.55 | 1.4 | 32 ms | 6.432 s | 550 nm |
| Fixation | 0.55 | 1.4 | 35 ms | 7.035 s | 550 nm |
| Stem differentiation | 0.55 | 1.4 | 35 ms | 7.035 s | 550 nm |
| UV irradiation | 0.55 | 1.4 | 35 ms | 7.035 s | 550 nm |
| High-temporal resolution system | 0.55 | 1.49 | 1.2 ms | 240 ms | 532 nm |

A table of the Illumination numerical aperture (NA), collection NA, exposure time, acquisition time, and imaging wavelength used for each experiment

a. In order to generate diffusion maps, $D$ must be calculated at each pixel so the natural log of the ACF cannot be averaged. To account for the increased noise, the ACF is filtered spatial with a Gaussian filter.

(9) The slope of the natural log of the ACF is calculated from the first two points to acquire the decay coefficient of the mean ACF.

(10) D is calculated from the decay coefficient using the following formula.

$$D = -\frac{\text{decay coefficent}}{4k^2}$$

**Statistical analysis**. All p-values are from two-tailed, heteroscedastic Student's t-tests unless otherwise noted as paired. All error bars are standard error of the mean (SEM). The coefficient of determination is reported as $R^2$ and is calculated from a linear regression as the square of the correlation coefficient. All n and p-values are reported in the text near the relevant comparison. All the parameters were calculated using Microsoft Excel (Microsoft Corporation, Redmond, WA).

**Cell culture**. All experiments (except for stem cell experiments) were performed with HeLa cells (ATCC, CCL-2). HeLa cells were grown in Gibco® formulated RPMI-1640 Media (Life Technologies, Carlsbad California) supplemented with 10% FBS (Sigma Aldrich, St. Louis Missouri) and grown at 37ºC and 5% $CO_2$. All of the cells in this study were maintained between passage 5 and 20. Cell were grown in 50 mm petri dishes with uncoated size 0 or 1 glass coverslip bottoms (MatTek, Ashland Massachusetts). Petri dishes were seeded with between 10,000 and 50,000 cells in 2 ml of the cell appropriate media at the time of passage. Cells were allowed at least 24 h to re-adhere and recover from trypsin-induced detachment. Imaging was performed when the surface confluence of the slide was between 40–70%.

**Stem cell culture and differentiation**. Human mesenchymal stem cells (hMSCs, ATCC, PCS-500-012) were cultured in Dulbecco's Modified Eagle Medium (DMEM), with 4.5 g per L glucose, and supplemented with 10% FBS and 5 ml 10x penicillin-streptomycin. For differentiation studies, cells were plated $1.5 \times 10^4$ cell per ml in 24-well glass-bottom PWS plates. After 2 days, regular DMEM was switched to hMSC Osteogenic Differentiation Medium (Lonza) containing B-glycerophosphate, ascorbate and dexamethasone. Media was changed every other day and cells were imaged on day 4 post-induction. To confirm differentiation, ALP colorimetric assay was performed on fixed hMSCs after imaging live with dual-PWS.

**Alexa fluor 488 phalloidin experiments**. Cells were irradiated with UV for ~12 min. First, cells were washed with PBS. Then, cells were fixed with 3.6% paraformaldehyde solution for ~10 min. A permeabilization/blocking step was performed with 0.1% Triton X-100, 1% bovine serum albumin in PBS for 5 min. Finally, cells were stained for 20 min with 40x dilution Alexa Fluor 488 Phalloidin (Thermo Fisher Scientific). Fluorescence quantification is performed as follows. Regions of interest are created for each whole cell. Raw fluorescence values are averaged within the ROI and normalized to the fluorescence values of cells that have undergone the cellular paroxysm.

**Annexin V experiments**. Cells were irradiated with UV for ~12 min. Then, 10% of the media was removed and replaced with an equivalent volume of 10X Annexin V Binding Buffer. Next, cells were stained with ~100x dilution of Annexin V-FITC Conjugate for 10 min. For some runs, staining was performed before the UV irradiation, so that multiple runs could be accomplished within the same dish. Annexin V Binding Buffer and Annexin V-FITC Conjugate come from Annexin V-FITC Early Apoptosis Detection Kit from Cell Signaling Technologies. Annexin V status was determined by utilizing a threshold of intensity and assigning a binary relationship using visual examination of the fluorescence images.

**Propidium iodide experiments**. Cells were irradiated with UV for ~12 min. Then, 10% of the media was removed and replaced with an equivalent volume of 10X Annexin V Binding Buffer. Next, cells were stained with ~9x Propidium Iodide for 10 min. For some runs, staining was performed before the UV irradiation. Annexin

V Binding Buffer and Propidium Iodide came from Annexin V-FITC Early Apoptosis Detection Kit from Cell Signaling Technologies. PI status was determined by utilizing a threshold of intensity and assigning a binary relationship using visual examination of the fluorescence images.

**BrdU experiments**. Cells were irradiated with UV for ~15 min. Then, BrdU (Cell Signaling Technology Bu20a; Catalog Number: 5292) was added to a concentration of 0.03 mg per mL and incubated at 37 C for 30 min. The cells were then fixed and stained with a secondary antibody at 1000x dilution (Invitrogen Alexa Fluor 532 Goat Anti-mouse IgG, Catalog Number: A-11002) according to the manufacturer's protocol found at https://www.cellsignal.com/contents/resources-protocols/immunofluorescence-protocol-for-labeling-with-brdu-antibody/5292-if. BrdU incorporation was determined by utilizing a threshold of intensity and assigning a binary relationship using visual examination of the fluorescence images.

**Caspase-3/7 and MitoTracker experiments**. Baseline measurements were acquired before UV irradiation. Cells were irradiated with UV for 1, 3, 6, and 20 min. After 2.5 or 20 h, cells were stained with 100 nM MitoTracker Orange CMTMRos and 4 µM CellEvent Caspase-3/7 Green Detection Reagent in RPMI media. After 30 min of incubation (37 C), the stained media was washed out and replaced with regular RPMI media. Dual-PWS and fluorescence images were acquired at this point.

## Data availability
All analyzed data and data statistics supporting the findings of this study are contained within the manuscript and its Supporting Information files. Raw data are available from the corresponding author upon reasonable request.

## Code availability
Computer code that support the findings of this study are available from the corresponding author upon reasonable request.

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

## Acknowledgements
This work was supported by fellowships and grants from the National Institutes of Health grant R01-GM105847, R01 CA200064, R33CA225323, 1R01CA228272, R01CA225002, R01EB016983, R01CA165309, K99 GM123195, the National Cancer Institute grant U54-CA193419 (CR-PS-OC), the NIH Chemistry of Life Processes Pre-doctoral Training grant #5T32GM105538-03 and the National Science Foundation grant CBET-1240416. An award of computer time was provided by the INCITE program. This research used resources of the Argonne Leadership Computing Facility, which is a DOE Office of Science User Facility supported under Contract DE-AC02-06CH11357.

## Author contributions
S.G. designed the study, performed experiments, analyzed data, developed the theory, validated the system, and wrote the manuscript. L.M.A. designed the study, performed experiments, analyzed data, and wrote the manuscript. L.C. developed the theory and reviewed the manuscript. A.E. performed experiments and reviewed the manuscript. A.E., D.Z. and W.W. validated the system and reviewed the manuscript. G.M.B. and S.M. performed experiments, analyzed data, and reviewed the manuscript. J.E.C., A.D.S., H.S., J.F.M., G.A.A. and I.S. designed the study and reviewed the manuscript. V.B. designed the study, developed the theory, and reviewed the manuscript.
