## [Peer Review File · Nature Communications]

Reviewers' comments:

Reviewer #1 (Remarks to the Author):

This is an interesting manuscript, reporting on an interferometric method to study cell dynamics. I believe that, after placing the work in the proper context of previous publications and straightening out a few confusions from the theoretical description, this paper will be well received.

1. The authors seem unaware of a large body of work in quantitative phase imaging applied to cell dynamics (see, for example, a recent review of QPI: Park et al., Nat. Phot, 2018 or the book from 2011). There are several approaches that should be discussed in context here:

- a. Nanoscale cell membrane fluctuations have been studied interferometrically for decades
- b. Dispersion-relation phase spectroscopy is a generalized form of dynamic light scattering, with spatially resolved information. The information on diffusion coefficients and deterministic transport is extracted from QPI data in the space-time frequency domain (via the dispersion relation). This seems to be most relevant here and may provide a way to simplify the current calculations.
- c. Phase correlation imaging is related to the above, but the images are mapped in terms of correlation times, also providing diffusion coefficients.
- d. Magnified image spatial spectrum (MISS) microscopy has been shown to extract diffusion coefficients associated with nanoparticles down to 20 nm.

2. It would be useful to the reader to see an experimental setup and understand exactly what is measured.

3. k seems to be the conjugate variable to z , according to Eq. 1 in the supplemental. Thus, it is incorrect to maximize it at $k \cdot NA$, because this limit applies to the transverse wavevector magnitude. Moreover, Eq. 4 shows an equation that depends on k , even though it was integrated over k^3 . Furthermore, the same k seems to occur in the correlation time. However, this spatial frequency has to be 2D. The $4Dk^2$ quantity should be $4Dk_{\text{perp}}^2$.

4. The authors may know that the interpretation of the backscattered field and how it relates to the sample's dry mass, is somewhat subtle, when one considers the transverse k -vectors. This interpretation of the phase map for a backscattered field has been calculated recently, under the first order Born approximation [1]. So the 1D approximation shown in the supplemental is not needed. Note that the measurements are (x, y, t) , i.e., spatially 2D.

5. The sensitivity of the instrument can be measured directly by performing the Diffusion coefficient extraction on static samples, such as fixed cells.

6. In Fig. 1, why not map the correlation time, like in ref. b above? In other words, why not calculate the second order moment of the autocorrelation function. The static backscattered intensity is prone to errors due to underestimations explained in point 3 above. Also, this quantity (σ_t) seems sensitive to the depth of field as well as the point spread function. The authors should discuss these dependences and how they were managed. Figure 1 needs color bars and units.

7. The authors should discuss whether the method can distinguish between random and deterministic transport.

8. For measuring single cells, it seems that a transmission geometry provides a more straight forward interpretation of the data. Is there some advantage in using backscattering?

9. I was not able to fully judge the new phenomenon presented here, as the second supplemental movie was not available in my download.

References

1. C. Hu and G. Popescu, Physical significance of backscattering phase measurements, Optics Letters, 42, 4643-4646 (2017).

Reviewer #2 (Remarks to the Author):

The work by Gladstein et al uses multimodal partial wave spectroscopy to study nanostructure and macromolecular dynamics inside cells at high spatial and temporal resolution. The dynamics of mass and structural heterogeneity inside cells is studied during cell fixation and stem cell differentiation. This manuscript also describes macromolecular dynamics inside of cells during UV irradiation and notes a spontaneous burst of motion here called cell paroxysm. The ability to observe processes lasting only milliseconds has been enabled by using the newly developed technique. The major novelty of this manuscript is the description of a modified form of partial wave spectroscopy (PWS) with improved temporal resolution. Previous work used PWS to study nanostructures in cells. In the present work acquisition of 3D time cubes has enabled the resolution of macromolecular dynamics in milliseconds. Overall, the technical description is generally sound but could use refinement. The description of biological phenomena should be improved in a major revision.

Specific concerns:

1. More description of the "cellular paroxysm" would be helpful to clarify the claim of discovery. From the results it is unclear what this paroxysm is, or if it is just an artifact at isolated pixels.
 - The video 2 mentioned in the text is unavailable. This movie is essential to a main biological finding of the manuscript.
 - Is this motion visible in other features of the cell (ex. larger scale motions)?
 - What is the timing of this event relative to PS externalization? The claim is that it is related to PS externalization, but the timing of this relationship would be instructive
 - The methods used for quantification of fluorescence (ex. Annexin V) is not well described.
2. The text mentions that there is an increase in noise with acquisition time, but the reason and limitations of the low SNR should be discussed as this may have a major impact on the applicability of the results.
3. The improvement in temporal resolution is a main novelty of the work and thus needs a better description. A figure showing the components of this technique, and how it differs from the group's previous work, would be useful and help with any future work building on this approach.
4. The principle of this technique, particularly the faster approach presented here using a single wavelength, seems similar to DLS. How is the current technique different or advantageous compared to DLS?
5. The stem cell differentiation section seems out of place in this manuscript. This may fit in a paper describing multiple applications of the approach, but the manuscript seems to focus on analyzing motion during cell death. At the very least, the results for stem cell differentiation should be compared to the expected results and how they are relevant biologically. There is no explanation of results in this section.
6. The description of motion spectra of cells before and during fixation (lines 142-146) would be easier to interpret if referenced to a figure. I understand this was likely not referenced to figure 2 to keep the figure order correct, but in this case, this section should be moved to the corresponding results section.
7. The statistical analysis is generally appropriate, but it would be helpful if p-values were indicated on the figures.

Reviewer #3 (Remarks to the Author):

The authors present a multimodal label-free interferometric imaging platform for measuring intracellular nanoscale structure and macromolecular dynamics in living cells. Applying this system

in vitro, they explore changes in higher-order chromatin structure and dynamics that occur due to cellular fixation, stem cell differentiation, and ultraviolet (UV) light irradiation. Finally, they discover a new phenomenon, cellular paroxysm, a near instantaneous, synchronous burst of motion that occurs early in the process of UV induced cell death.

During apoptosis, actin depolymerization is associated with caspase activation. That means that extensive degradation of molecules is also concomitant with cytoskeleton reorganization. In genuine apoptosis this process is accompanied by the formation of the apoptotic microtubule network and slight plasma membrane permeabilization.

Did the authors check if UV-induced apoptosis in this particular cell line is genuine apoptosis? Later when apoptotic microtubules depolymerized apoptotic cells undergo secondary necrosis and cell membrane become permeable for PI.

To me is not clear in which moment the authors detect the cellular paroxysm: during genuine apoptosis or when the cells undergo secondary necrosis.

The authors claim that "this burst of motion is connected to a disruption of the cytoskeletal and cell membrane structures" but during genuine apoptosis the plasma membrane is intact (PI negative, annexin V positive).

In Figure 7, plasma membrane is disrupted (PI positive) in UV irradiated cells. This process is accompanied by minimal incorporation of BrdU into the nuclei without cellular paroxysm. In these circumstances cells should be in secondary necrosis, however the nucleus is not fragmented suggesting that the cells after UV irradiation die by primary necrosis.

My impression is that UV-induced apoptosis needs better characterization.

Figure 6. What do MB and NMB stand for?

We want to thank all the reviewers for their helpful feedback and suggestions. We feel that the manuscript has been significantly improved.

Reviewers' comments:

Reviewer #1 (Remarks to the Author):

This is an interesting manuscript, reporting on an interferometric method to study cell dynamics. I believe that, after placing the work in the proper context of previous publications and straightening out a few confusions from the theoretical description, this paper will be well received.

1. The authors seem unaware of a large body of work in quantitative phase imaging applied to cell dynamics (see, for example, a recent review of QPI: Park et al., Nat. Phot, 2018 or the book from 2011). There are several approaches that should be discussed in context here:

- a. Nanoscale cell membrane fluctuations have been studied interferometrically for decades**
- b. Dispersion-relation phase spectroscopy is a generalized form of dynamic light scattering, with spatially resolved information. The information on diffusion coefficients and deterministic transport is extracted from QPI data in the space-time frequency domain (via the dispersion relation). This seems to be most relevant here and may provide a way to simplify the current calculations.**
- c. Phase correlation imaging is related to the above, but the images are mapped in terms of correlation times, also providing diffusion coefficients.**
- d. Magnified image spatial spectrum (MISS) microscopy has been shown to extract diffusion coefficients associated with nanoparticles down to 20 nm.**

We thank the reviewer for the suggestion, and agree that dual-PWS should be introduced within the context of QPI based dynamics techniques. We have added references and a brief discussion of these techniques to the manuscript introduction.

From the introduction, after a brief discussion of the issues with label based dynamics techniques:

“To address these issues, techniques have been developed based on quantitative phase imaging (QPI) to measure intracellular dynamics without the use of labels¹. Techniques such as phase correlation imaging², magnified image spatial spectrum microscopy³, and dispersion-relation phase spectroscopy⁴ extract diffusion coefficients from temporal fluctuations in phase via the dispersion relation. These techniques have led to interesting biological discoveries, such as a universal behavior where intracellular transport is diffusive at small scales and deterministic at large scales as well as differences in molecular motion between senescent and quiescent cells.

Building upon these advancements, we present a novel label-free interferometric platform (dual-PWS) that captures the temporal behavior and structural organization of macromolecular assemblies in live cells.”

2. It would be useful to the reader to see an experimental setup and understand exactly what is measured.

We thank the reviewer for this very helpful suggestion; it is very important to us that the reader understands what we are measuring. We have created a new figure (Figure 1) that contains a schematic of the Dual-PWS instrumentation and diagrams clarifying the acquisition process. Additionally, we have modified the Microscope Design and Data Acquisition Methods section to clarify and provide additional details on the system design, acquisition, and analysis process.

“Microscope Design and Data Acquisition

The dual-PWS system consists of a commercial microscope base (Leica DMIRB) equipped with a broad-spectrum white light LED source (X-Cite 120LED), 63x oil immersion objective (Leica HCX PL APO, NA 1.4 or 0.6), spectral filter (CRi VariSpec LCTF), and CCD camera (Hamamatsu Image-EM CCD). Light from the source is focused onto the sample with an illumination NA of 0.55 and the backscattered light is collected by passing it through the spectral filter and imaged at the CCD (**Figure 1a**). Multiple wide-field monochromatic images are obtained for each acquisition and stored in a three-dimensional image cube as described below.

Structural measurements (Σ_s) are collected by acquiring multiple backscattered wide-field monochromatic images across a range of wavelengths (500-700nm) to produce a three-dimensional image cube, $I(\lambda, x, y)$, where λ is the wavelength and (x, y) correspond to pixel positions (**Figure 1b**). Each frame within a spectral data cube is acquired with a 35ms exposure, and the total cube can be obtained in under two seconds depending on the number of wavelengths collected⁵. As described above, Σ_s is extracted by calculating the standard deviation of the spectral interference at each pixel (x, y) . Similarly, dynamics measurements (Σ_t^2 , m_f , and D) are collected by acquiring multiple backscattered wide-field images at a single wavelength (550nm) over a period of time (acquisition time), to produce a three-dimensional image cube, $I(t, x, y)$, where t is time and (x, y) correspond to pixel positions (**Figure 1b**). D is extracted by calculating the decay rate of the autocorrelation of the temporal interference and m_f is calculated by normalizing the variance of the temporal interference (Σ_t^2) at each pixel (x, y) ; Further details on these calculations are included in the **SI Temporal Interference Theory**. To correct for temporal lamp fluctuations (and other system noise), reference dynamics image cubes are collected from an empty field of view within the sample; Σ_t^2 calculated from the reference cube is subtracted from our sample's Σ_t^2 to remove noise contributions. For this study, temporal measurements consisted of at least 201 frames acquired with 1.2ms, 32ms, or 35ms exposures each for a total time of ranging between 240ms to 7 seconds. In theory, the exposure and acquisition time can be adjusted to match the biological phenomenon of interest (see **SI Timescale Sensitivity**). In the current configuration, the temporal and structural measurements are performed sequentially in <15 seconds. “

a) Dual-PWS Instrumentation

b) Dual-PWS Acquisition and Analysis

$$\Sigma_S$$

$$\Sigma_t^2 \propto m_f D$$

“Figure 1: Dual-PWS Instrumentation, Acquisition, and Analysis. (a) Schematic of the Dual-PWS instrumentation. Broadband white light from a Light-emitting diode (LED) is focused onto the sample. The backscattered light is spectrally filtered through a liquid crystal tunable filter (LCTF) and imaged with a CCD camera. (b) (left) Structural measurements (Σ_s) are collected by acquiring multiple backscattered wide-field monochromatic images across a range of wavelengths (500-700nm) to produce a three-dimensional image cube, $I(\lambda, x, y)$, where λ is the wavelength and (x, y) correspond to pixel. Σ_s is extracted by calculating the standard deviation of the spectral interference at each pixel (x, y) . (right) Dynamics measurements (Σ_t^2 , m_f , and D) are collected by acquiring multiple backscattered wide-field images at a single wavelength (550nm) over a period of time (acquisition time), to produce a three-dimensional image cube, $I(t, x, y)$, where t is time and (x, y) correspond to pixel positions. D is extracted by calculating the decay rate of the autocorrelation of the temporal interference and m_f is calculated by normalizing the variance of the temporal interference (Σ_t^2) at each pixel (x, y) .”

3. k seems to be the conjugate variable to z , according to Eq. 1 in the supplemental. Thus, it is incorrect to maximize it at $k \cdot NA$, because this limit applies to the transverse wavevector magnitude. Moreover, Eq. 4 shows an equation that depends on k , even though it was integrated over k^3 . Furthermore, the same k seems to occur in the correlation time. However, this spatial frequency has to be $2D$. The $4Dk^2$ quantity should be $4Dk_{\text{perp}}^2$.

We apologize for the confusion and have added additional details to the **SI Temporal Interference Theory** to hopefully clear this up. k referred to in Eq.1 is the scalar wavenumber of the illumination light, which originated in the derivations from k_z the conjugate variable to z [1]. \mathbf{k} (bold) in equation 4 is the 3D frequency-space wavevector. The Fourier transform of our ACF is integrated over a disk with radius kNA centered at $k_z=2k$; this should be the correct limit for both the transverse and z wavevectors. k in the correlation time is again the scalar wavenumber of the illumination light. This originates from the magnitude ($2k$) of the 3D spatial frequency vector \mathbf{k} due to the 3D spatial Fourier transform in eq. 14.

1. Cherkezyan, L. *et al.* Interferometric Spectroscopy of Scattered Light Can Quantify the Statistics of Subdiffractive Refractive-Index Fluctuations. *Physical Review Letters* **111**, 033903 (2013).

Some of the relevant modifications to the **SI Temporal Interference Theory**:

After eq. 1

“where k is the scalar wavenumber of the illumination light”

After eq. 4

“where \mathbf{k} is the 3D frequency-space wave vector, $T_{3D} = T_{kNA} T_{ks}$ a disk with radius kNA in frequency-space centered at $k_z = 2k$.”

After eq. 13

“The 3D spatial Fourier transform of equation 14 is”

4. The authors may know that the interpretation of the backscattered field and how it relates to the sample's dry mass, is somewhat subtle, when one considers the transverse k -vectors. This

interpretation of the phase map for a backscattered field has been calculated recently, under the first order Born approximation [1]. So the 1D approximation shown in the supplemental is not needed. Note that the measurements are (x, y, t) , i.e., spatially 2D.

Equation 1 in the **SI Temporal Interference Theory** is not a 1D approximation of a 3D problem. It has previously been shown that to describe a microscope-generated spectrum (a 1-D signal), the 3-D problem of light propagation is reduced to a 1-D problem where the RI is convolved with the Airy disk in the transverse plane [1]. The reviewer is correct that the measurements are spatially 2D; we were essentially calculating the spatially expected value of our markers as $n\Delta(r)$ is random. We have modified the theory to start at 2D spatial measurements (x, y, k, t) and clearly state when we take the expected value to calculate the sample statistics.

1. Cherkezyan, L. *et al.* Interferometric Spectroscopy of Scattered Light Can Quantify the Statistics of Subdiffractive Refractive-Index Fluctuations. *Physical Review Letters* **111**, 033903 (2013).

5. The sensitivity of the instrument can be measured directly by performing the Diffusion coefficient extraction on static samples, such as fixed cells.

We thank the reviewer for this suggestion. We agree that this is a helpful method to understand the experimental limits of the system and this analysis has been added to the manuscript in SI section **Timescale Sensitivity**.

From SI Timescale Sensitivity Section

“Experimentally, the instrument sensitivity can be measured by extracting dynamics parameters from a static sample. Measuring m_f from static samples provides a lower limit of detectable motion. Note that background subtraction is applied in regular m_f calculations, so motion below our sensitivity limit will be close to zero instead of the sensitivity level. Without background subtraction, m_f measurements of fixed HeLa cells provide a sensitivity limit of $m_f = 6.03e-20 \pm 1e-21$ g ($n = 7$). Measuring diffusion coefficients from static samples provides an upper limit to our temporal sensitivity assuming the system noise is relatively uncorrelated, resulting in the fastest detectable correlation decay. Similar to theoretical limit of $0.065 \frac{\mu M^2}{sec}$ measured through simulations, an experimental limit of $D = 0.08 \pm 0.001 \frac{\mu M^2}{sec}$ ($n = 7$) was measured using fixed HeLa cells. For this analysis, noise removal, such as background subtraction and removal of pixels with low SNR are not performed as we are trying to measure system noise.”

6. In Fig. 1, why not map the correlation time, like in ref. b above? In other words, why not calculate the second order moment of the autocorrelation function. The static backscattered intensity is prone to errors due to underestimations explained in point 3 above. Also, this quantity (σ_t) seems sensitive to the depth of field as well as the point spread function. The authors should discuss these dependences and how they were managed. Figure 1 needs color bars and units.

We have added maps of diffusion coefficient as well as color bars and units to Figure 1 (now referred to as Figure 2 in the revised manuscript). We have used diffusion coefficient instead of correlation time to be consistent with how the rate of molecular motion is represented in this manuscript. Diffusion coefficient is proportional to correlation time, $D=1/4k^2t_c$. A discussion of the dependencies mentioned above were added to the **SI Temporal Interference Theory** section.

“It should be noted that Σ_t is sensitive to instrument parameters such as depth of field, substrate refractive index, etc. For biological measurements, it is important to use parameters such as m_f , where these dependencies are removed through normalization with the proper prefactor calculated above. Additionally, the backscattered intensity is prone to errors along the transverse direction⁶. Due to these variations, these parameters are most accurate after calculating the expected values over (x', y') .”

“Figure 2: Nanosphere Phantoms. **a)** Representative \sum_t maps of polystyrene nanosphere phantoms created with a variety of nanosphere sizes (25nm, 37.5nm, 50nm, and 100nm) and concentrations (0.1% and 0.3% volume fractions). **b)** Bar graph with individual data points comparing \sum_t for the phantoms represented in (a) with analytical theory. The PWS measurements of these phantoms match well with theory ($R^2=0.78$) and show that \sum_t is sensitive to the volume fraction of spheres (ϕ) and the mass of the moving spheres (m_c). **c)** Representative m_f maps of 0.1% volume fraction polystyrene nanosphere phantoms created with a variety of nanosphere sizes (25nm, 37.5nm, 50nm, and 100nm) **d)** Bar graph with individual data points comparing experimentally measured D with theoretical values calculated using the Stokes-Einstein equation for 0.1% sphere phantoms with sizes 25nm, 37.5nm, 50nm, and 100nm. The PWS measured D values closely match the predicted values for a correlation of $R^2 = 0.99$. Scale bar is 8 μ M.”

7. The authors should discuss whether the method can distinguish between random and deterministic transport.

Theoretically, dual-PWS should be able to distinguish between random and deterministic transport, but it will require some hardware modifications to properly perform these measurements. This could be accomplished by substituting the advection-diffusion equation into the temporal interference theory instead of the general diffusion equation. This would result in a normalized autocorrelation decay solution that depends on both the diffusion coefficient (D) and the advection velocity (v).

$$\frac{B_{\delta I}(\tau, k)}{B_{\delta I}(0, k)} = e^{-2kv\tau - 4k^2 D\tau}$$

Next, to decouple the random and deterministic components, dual-PWS dynamics measurements will need to be acquired with multiple wavelengths simultaneously. This could be accomplished by replacing the LCTF (and CCD) with a hyperspectral camera, multichannel AOTF, or other spectral filter that allows multiple wavelengths to be acquired at the same time. Thank you for bring up this topic, this would be a great modification for the next version of the dual-PWS system.

A brief mention of this has been added to the main text:

“Future modifications to the dual-PWS instrumentation may enable decoupling of diffusive and deterministic transport by utilizing hyperspectral imaging hardware⁴.”

8. For measuring single cells, it seems that a transmission geometry provides a more straight forward interpretation of the data. Is there some advantage in using backscattering?

We agree with the reviewer that the transmission geometry provides a more straight forward interpretation of the phase, which is essential for QPI based imaging techniques, but the backscattering geometry is quite important for the lengthscale sensitivity of PWS. PWS is sensitive to nanoscale structural organization and macromolecular dynamics by measuring the power spectral density of refractive index variations over spatial frequencies within T_{3D} . Under a forward scattering geometry, T_{3D}

is shifted to be centered at $k_z = 0$ instead of $k_z = 2k$; additionally, the k_z range of spectral PWS measurements would become very narrow. These new T_{3D} s would contain mostly low spatial frequencies significantly reducing the sensitivity of PWS to nanoscale information. Not only does T_{3D} include higher spatial frequencies in the backscattering geometry, but it does not include zero and low longitudinal spatial frequencies, reducing PWS sensitivity to large structures. Both of these advantages would be lost in the transmission geometry. Additionally, the reference wave would become orders of magnitude larger in the forward scattering geometry. While this would amplify the spectral/temporal variations in our signal, the DC component of the signal would increase significantly more than the variable portion, effectively decreasing the SNR of the PWS signal.

9. I was not able to fully judge the new phenomenon presented here, as the second supplemental movie was not available in my download.

We apologize for this error. We are uncertain why this video was unavailable, but we will confirm that it is properly uploaded and included in this revised submission.

References

1. C. Hu and G. Popescu, Physical significance of backscattering phase measurements, *Optics Letters*, **42**, 4643-4646 (2017).

Reviewer #2 (Remarks to the Author):

The work by Gladstein.et.al uses multimodal partial wave spectroscopy to study nanostructure and macromolecular dynamics inside cells at high spatial and temporal resolution. The dynamics of mass and structural heterogeneity inside cells is studied during cell fixation and stem cell differentiation. This manuscript also describes macromolecular dynamics inside of cells during UV irradiation and notes a spontaneous burst of motion here called cell paroxysm. The ability to observe processes lasting only milliseconds has been enabled by using the newly developed technique. The major novelty of this manuscript is the description of a modified form of partial wave spectroscopy (PWS) with improved temporal resolution. Previous work used PWS to study nanostructures in cells. In the present work acquisition of 3D time cubes has enabled the resolution of macromolecular dynamics in milliseconds. Overall, the technical description is generally sound but could use refinement. The description of biological phenomena should be improved in a major revision.

Specific concerns:

1. More description of the "cellular paroxysm" would be helpful to clarify the claim of discovery. From the results it is unclear what this paroxysm is, or if it is just an artifact at isolated pixels.

We thank the reviewer for this suggestion. We have clarified the definition of the cellular paroxysm, added images of the event to figure 5, as well as a new supplemental section (**Characterization of the Cellular Paroxysm**) providing further characterization of the cellular paroxysm.

From the main text:

“...we examine the temporal responses of individual cells due to UV irradiation (**Figure 5cd**). In particular, a near-instantaneous, large scale burst of motion (transient increase in m_f across the whole cell), termed a cellular paroxysm, occurs between 10 and 20 minutes that is predominantly asynchronous from cell to cell (**Figure 6a; Video 1 between 00:04 – 00:06**).”

The evidence indicates that the cellular paroxysm is a dynamic biological response, not an artifact at isolated pixels:

- 1) This change in fractional moving mass only occurs within the cells, not in the background of the mf images.
- 2) The cellular paroxysm only occurs in response to a cellular perturbation (UV irradiation). It has not observed in control measurements.
- 3) The cellular paroxysms have been correlated to other biological measurements (phosphatidylserine externalization and actin depolymerization).

Figure 6

a**b****c****d**
“Figure 6: Cellular Paroxysm Temporal Analysis. **(a)** Representative m_f maps of a cell undergoing the cellular paroxysm. **(b)** Example temporal spectra from a single pixel during a cellular paroxysm. Using spectral feature detection the timing of the event can be determined for each pixel reducing the temporal resolution from the full acquisition time (~ 7 s) to a single exposure time (35ms). **(c)** Example histogram of the timing of the event for each pixel within a single cell. **(d)** Map displaying the timing of the event for each pixel within a representative cell. The spatial distribution of this timing shows that the initial motion occurs simultaneously within a single 35ms frame on opposite ends of the cell.”

From the SI:

“Characterization of the Cellular Paroxysm

In general, the cellular paroxysm is characterized as a near-instantaneous, large scale burst of motion (transient increase in m_f across the whole cell) that is asynchronous from cell to cell. Individually, there can be variations in timing, localization, and synchronicity during the cellular paroxysm. To better characterize these variations in the paroxysm, we have performed a subsampling analysis to increase the temporal resolution of our dynamics measurements, effectively decreasing the acquisition time from ~ 7 seconds to 0.7 seconds (details below). While this analysis doesn't improve the temporal resolution as much as the single frame analysis used in **Figure 5bcd** and **Video 2**, it allow us to output m_f maps, which the single frame analysis cannot, and identify features that are not clearly identified in the timing of cellular paroxysm maps (**Figure 5d**). Through this analysis, we have identified some interesting variations from the typical cellular paroxysm. Most cellular paroxysms occur across the entire cell, but occasionally we have observed partial paroxysms that only occur in one region of the cell (**SI Figure 6**). In generally, the paroxysm initiates across the entire cell instantaneously, but we have observed a few cells that exhibit a wavelike initiation, where the paroxysm starts in one region of the cell and spreads to the rest of the cell over time (**SI Figure 7**). Typically, paroxysms are asynchronous from cell to cell across the field of view, but sometimes adjacent cells synchronize and the paroxysms initiate simultaneously (**SI Figure 8**). While the three previous features are relatively rare, cyclical paroxysms, where a smaller secondary burst of motion occurs after the initial paroxysm, appear regularly (**SI Figure 9**). These secondary paroxysms can appear milliseconds to tens of seconds after the initial paroxysm. This analysis has shown that a significant amount of individual variation can occur within the cellular paroxysm phenomenon.

Subsampling Analysis Method

To increase the temporal resolution in post-processing, a single dynamics data cube is split along the time axis into ten separate data cubes reducing the acquisition time from 7.035 to 0.7035 seconds. Each of these reduced data cubes are analyzed using the standard algorithm to produce m_f maps. This reduction in acquisition time increases noise in the m_f maps. A 2-D median spatial filter using a 3-by-3 super-pixel is applied to the maps to reduce this spatial noise.”

SI Figure 6

Partial Cellular Paroxysms

“SI Figure 6: Partial Cellular Paroxysms. Partial paroxysms are one of the features identified during the characterization of cellular paroxysms. Generally, paroxysms occur within the entirety of the cell, but occasionally, they will only effect a small section of the cell.”

SI Figure 7

Wavelike Initiation of Cellular Paroxysms

“SI Figure 7: Wavelike Initiation of Cellular Paroxysms. Wavelike initiation of the cellular paroxysm is one of the features identified during the characterization of cellular paroxysms. Generally, paroxysms initiate across the entirety of the cell instantaneously, but occasionally, they will initiate in one part of the cell and spread out in a wavelike manner across the entire cell.”

SI Figure 8

Synchronous Cellular Paroxysms of Adjacent Cells

“SI Figure 8: Synchronous Cellular Paroxysms. Synchronous cellular paroxysms are one of the features identified during the characterization of cellular paroxysm. Generally, paroxysms initiate asynchronously from cell to cell, but occasionally, adjacent cells (generally in contact with each other) will undergo cellular paroxysm simultaneously.”

SI Figure 9

Cyclical Cellular Paroxysms

“SI Figure 9: Cyclical Cellular Paroxysms. Cyclical cellular paroxysms are one of the features identified during the characterization of cellular paroxysm. Sometimes, after the initial cellular paroxysm, a secondary smaller paroxysm will occur milliseconds to tens of seconds after the initial paroxysm has subdued.”

- The video 2 mentioned in the text is unavailable. This movie is essential to a main biological finding of the manuscript.

We apologize for this error. We are uncertain why this video was unavailable, but we will confirm that it is properly uploaded and included in this revised submission.

- Is this motion visible in other features of the cell (ex. larger scale motions)?

The cellular paroxysm is generally a large scale motion that is visible across the entire cell utilizing the dual-PWS system; images of the paroxysm have been added to **Figure 5a** to clarify this. On rare occasions, the motion is only observed in smaller regions of the cell; this has been characterized in SI section **Cellular Paroxysm Characterization** (also included above in the response to your first question).

- What is the timing of this event relative to PS externalization? The claim is that it is related to PS externalization, but the timing of this relationship would be instructive

Based on our experiments, cellular paroxysm and PS externalization occur at the same time (within seconds). This is observed by UV irradiating the cells for an intermediate duration in order to differentially induce the response in some, but not all cells within the field of view, before treating with Annexin V. In general, the cells that exhibited cellular paroxysm display positive Annexin V (PS externalization), while cell that experienced the same UV treatment, but were seconds to a couple minutes from undergoing the paroxysm display negative Annexin V.

Unfortunately, due to limitations in the Annexin V staining and imaging procedure, the precision of this timing cannot be further increased. We cannot simultaneously image PS externalization and PWS dynamics; we have to complete the PWS imaging series before imaging the Annexin V.

Added to the main text:

“From these results, PS externalization and the cellular paroxysm occur either simultaneous or potentially PS externalization occurs a few seconds before. Due to limitations in the Annexin V staining and imaging procedure, the precision of this timing cannot be further increased as we have to complete the PWS imaging series before imaging the Annexin V.”

- The methods used for quantification of fluorescence (ex. Annexin V) is not well described.

We apologize for the lack of clarity in the description of the analysis of Annexin V experiment. Alexa Fluor 488 Phalloidin fluorescence images were the only fluorescence images where the intensity of the staining is quantified in this manuscript. Details have been added to the methods section to clarify the analysis.

“Regions of interest are created for each whole cell. Raw fluorescence values are averaged within the ROI and normalized to the fluorescence values of cells that have undergone the cellular paroxysm.”

Positive/negative Annexin V status (as well as PI and BrdU status) were determined by utilizing a threshold of intensity and assigning a binary relationship using visual examination of the fluorescence images. To confirm that this original selection was not biased, we are including quantified results below. As with Phalloidin, whole cell ROIs were created and averaged for each cell. In this analysis, the fluorescence is normalized by the background fluorescence in each image to account for image to image variations. After quantification, it is clear that there is a complete separation in fluorescence intensity for cells that were identified as Annexin positive compared with Annexin negative, confirming that the original visual analysis was not biased.

Fluorescence experiments determined by visual examination added to the methods section.

“PI status was determined by utilizing a threshold of intensity and assigning a binary relationship using visual examination of the fluorescence images.”

“BrdU incorporation was determined by utilizing a threshold of intensity and assigning a binary relationship using visual examination of the fluorescence images.”

“Annexin V status was determined by utilizing a threshold of intensity and assigning a binary relationship using visual examination of the fluorescence images.”

2. The text mentions that there is an increase in noise with acquisition time, but the reason and limitations of the low SNR should be discussed as this may have a major impact on the applicability of the results.

We apologize for the confusion. Increased acquisition time will not increase noise, but will affect the temporal sensitivity, which is discussed in the SI under **Timescale Sensitivity** section. We have added additional information in the SI on the factors that influence SNR under **Dual-PWS Signal to Noise**.

“Dual-PWS Signal to Noise

Signal to Noise ratio (SNR) is an important consideration when acquiring dual-PWS images. Motion that produces a signal near or below the noise limit cannot be quantified. It should be noted the measurements of diffusion are particularly sensitive and will become inaccurate with low SNR. In general, SNR will be affected by the same parameters as any microscopy technique, but there are some special considerations that should be considered.

Exposure time: SNR can be directly increased by increasing exposure time, but increased exposure time will reduce sensitivity to faster processes (see SI Timescale Sensitivity).

Lamp Intensity: SNR can be directly increased by increasing the input lamp intensity. Theoretically, this will not affect the measurements in any other way, but high intensities of light can cause biological changes in cells (especially in the UV range). One potential solution is illumination side spectral filtering, which can reduce light intensity impacting the cells.

Camera Sensitivity and Noise: Improved camera sensitivity and reduced camera noise will improve SNR without any negative effects.

Cell Motion: The temporal interference signal originates from intracellular macromolecular dynamics. When comparing different cell lines or perturbations, increases or decreases in macromolecular dynamics will directly affect the SNR. As this is the parameter we are trying to measure, it cannot be tweaked to improve SNR, but it should be considered when planning and interpreting dual-PWS experiments.”

3. The improvement in temporal resolution is a main novelty of the work and thus needs a better description. A figure showing the components of this technique, and how it differs from the group's previous work, would be useful and help with any future work building on this approach.

We thank the reviewer for this suggestion; it is very important to us that the reader understands what we are measuring. First to clarify, the novelty of this work is to provide an entirely new modality for measuring macromolecular motion compared to previous versions of PWS that only measured structural organization. While the previous version of PWS could measure and quantify structural organization at

multiple time points in live cells, this would provide significantly less information on cellular dynamics than dual-PWS, which directly measures multiple parameters of intracellular motion.

To clarify the components of dual-PWS as well as the differences between the structural and dynamics measurements, we have created a new figure (**Figure 1**) that contains a schematic of the Dual-PWS instrumentation and diagrams clarifying the acquisition process. Additionally, we have modified the Microscope Design and Data Acquisition Methods section to clarify and provide additional details on the system design, acquisition, and analysis process.

“Microscope Design and Data Acquisition

The dual-PWS system consists of a commercial microscope base (Leica DMIRB) equipped with a broad-spectrum white light LED source (X-Cite 120LED), 63x oil immersion objective (Leica HCX PL APO, NA 1.4 or 0.6), spectral filter (CRi VariSpec LCTF), and CCD camera (Hamamatsu Image-EM CCD). Light from the source is focused onto the sample with an illumination NA of 0.55 and the backscattered light is collected by passing it through the spectral filter and imaged at the CCD (**Figure 1a**). Multiple wide-field monochromatic images are obtained for each acquisition and stored in a three-dimensional image cube as described below. Structural measurements (Σ_s) are collected by acquiring multiple backscattered wide-field monochromatic images across a range of wavelengths (500-700nm) to produce a three-dimensional image cube, $I(\lambda, x, y)$, where λ is the wavelength and (x, y) correspond to pixel positions (**Figure 1b**). Each frame within a spectral data cube is acquired with a 35ms exposure, and the total cube can be obtained in under two seconds depending on the number of wavelengths collected⁵. As described above, Σ_s is extracted by calculating the standard deviation of the spectral interference at each pixel (x, y) . Similarly, dynamics measurements (Σ_t^2 , m_f , and D) are collected by acquiring multiple backscattered wide-field images at a single wavelength (550nm) over a period of time (acquisition time), to produce a three-dimensional image cube, $I(t, x, y)$, where t is time and (x, y) correspond to pixel positions (**Figure 1b**). D is extracted by calculating the decay rate of the autocorrelation of the temporal interference and m_f is calculated by normalizing the variance of the temporal interference (Σ_t^2) at each pixel (x, y) ; Further details on these calculations are included in the **SI Temporal Interference Theory**. To correct for temporal lamp fluctuations (and other system noise), reference dynamics image cubes are collected from an empty field of view within the sample; Σ_t^2 calculated from the reference cube is subtracted from our sample's Σ_t^2 to remove noise contributions. For this study, temporal measurements consisted of at least 201 frames acquired with 1.2ms, 32ms, or 35ms exposures each for a total time of ranging between 240ms to 7 seconds. In theory, the exposure and acquisition time can be adjusted to match the biological phenomenon of interest (see **SI Timescale Sensitivity**). In the current configuration, the temporal and structural measurements are performed sequentially in <15 seconds. “

a) Dual-PWS Instrumentation

b) Dual-PWS Acquisition and Analysis

$$\Sigma_S$$

$$\Sigma_t^2 \propto m_f D$$

“Figure 1: Dual-PWS Instrumentation, Acquisition, and Analysis. (a) Schematic of the Dual-PWS instrumentation. Broadband white light from a Light-emitting diode (LED) is focused onto the sample. The backscattered light is spectrally filtered through a liquid crystal tunable filter (LCTF) and imaged with a CCD camera. (b) (left) Structural measurements (Σ_s) are collected by acquiring multiple backscattered wide-field monochromatic images across a range of wavelengths (500-700nm) to produce a three-dimensional image cube, $I(\lambda, x, y)$, where λ is the wavelength and (x, y) correspond to pixel. Σ_s is extracted by calculating the standard deviation of the spectral interference at each pixel (x, y) . (right) Dynamics measurements (Σ_t^2 , m_f , and D) are collected by acquiring multiple backscattered wide-field images at a single wavelength (550nm) over a period of time (acquisition time), to produce a three-dimensional image cube, $I(t, x, y)$, where t is time and (x, y) correspond to pixel positions. D is extracted by calculating the decay rate of the autocorrelation of the temporal interference and m_f is calculated by normalizing the variance of the temporal interference (Σ_t^2) at each pixel (x, y) .”

4. The principle of this technique, particularly the faster approach presented here using a single wavelength, seems similar to DLS. How is the current technique different or advantageous compared to DLS?

There many differences/advantages comparing dual-PWS with DLS. First, DLS is generally used for non-spatially localized measurements which would typically require detachment and lysis of cells for molecular analysis. Conversely, dual-PWS is an imaging technique, which can measure macromolecular dynamics in individual cells as well as spatially resolve the dynamic measurements to specific loci within the cell, including to organelles such the nucleus. Second, while both techniques can measure diffusion coefficients, dual-PWS measures additional parameters that DLS cannot such as structural heterogeneity and fractional moving mass. It should be noted that there are some imaging techniques based on quantitative phase imaging that are related to DLS. A brief discussion of these techniques as well as the advantages of dual-PWS has been added to the introduction.

“To address these issues, techniques have been developed based on quantitative phase imaging (QPI) to measure intracellular dynamics without the use of labels¹. Techniques such as phase correlation imaging², magnified image spatial spectrum microscopy³, and dispersion-relation phase spectroscopy⁴ extract diffusion coefficients from temporal fluctuations in phase via the dispersion relation. These techniques have led to interesting biological discoveries, such as a universal behavior where the intracellular transport is diffusive at small scales and deterministic at large scales as well as differences in molecular motion between senescent and quiescent cells.

...

In addition to measuring diffusion coefficients (similar to other label-free techniques), dual-PWS extracts additional quantifications of motion such as the fractional moving mass, which provides a measurement of the volume fraction of moving structures and the mass of moving

structures. Beyond that, dual-PWS is sensitive to motion occurring on (or confined to) length scales smaller than the diffraction limit.”

5. The stem cell differentiation section seems out of place in this manuscript. This may fit in a paper describing multiple applications of the approach, but the manuscript seems to focus on analyzing motion during cell death. At the very least, the results for stem cell differentiation should be compared to the expected results and how they are relevant biologically. There is no explanation of results in this section.

We appreciate the reviewer’s point that the discussion of stem cells as initially presented was confusing. The focus of the manuscript is on the development and multiple applications (cellular fixation, stem cell differentiation, and the cellular response to UV irradiation) of the dual-PWS system. There was initially more analysis and discussion on the application of UV irradiation/cell death because of the novel phenomenon discovered. While we do believe that this section should be included in the manuscript, we understand your concerns. To address these issues, we have modified the stem cell section to include additional discussion of our expected results based on previous publications, the biological implications, as well as additional analysis of the data demonstrating that physical properties within the nucleus can be utilized to discriminate cellular state. With these additions, we believe the focus of the manuscript is clarified while presenting novel findings for the stem cell community.

“The process of cellular differentiation is among the most widely studied in molecular biology and has been demonstrated to result in changes in the organization of chromatin in fixed cells as well as the motility of transcription factors^{7,8}. Particularly, Dixon et al delineated biases in allelic gene expression in embryonic stem cells versus their differentiated progeny, however the mechanism and rationale for higher-order chromatin influence on stem cell differentiation is still a subject of study. Human mesenchymal stem cells (hMSCs) differentiate into different specialized cell types—including adipocytes, chondrocytes, and osteoblasts—due to the integration of a host of chemical and physical signals. Using fluorescence correlation spectroscopy, previous studies have observed slower diffusion of individual transcription factors in stem cells in lineages where their functions are essential⁷, but no one has studied large scale changes in motion due to differentiation.

We hypothesized that in hMSCs there would be increased chromatin heterogeneity and higher mobility due to the unique abilities of hMSCs to access genes for multiple phenotypes, given their high differentiation potential. Conversely, we hypothesized that differentiated cells (e.g. osteoblasts) would have lower heterogeneity and mobility as a result of their committed lineage in which reversion to a stem cell state is no longer likely. Consistent with previous studies, we observe that the hMSC and osteoblast populations have very different states of chromatin folding and nuclear dynamics (**Figure 4a**) as well as significant heterogeneity within each population (**SI Figure 18**). Specifically, hMSCs were observed to have a more heterogeneous chromatin structure [$\Delta\Sigma_s = 25.0 \pm 2.2\%$ (SEM), p-value = $9.6e-24$], a higher fractional moving mass [$\Delta m_f = 24.8 \pm 3.9\%$ (SEM), p-value = $8.8e-10$], and slower

macromolecular motion [$\Delta D = 26.7 \pm 3.4\%$ (SEM), p-value = $3.87e-11$] when compared to differentiated osteoblasts (n=166 hMSC and n=102 osteoblasts) (**Figure 4b**). These findings support our hypothesis that committed lineages such as osteoblasts lack the heterogeneity and molecular motion that potentiate stem cells as well as expand upon previous studies showing that the changes in individual transcription factor diffusion extends to large scale changes in nuclear diffusion. The biological implications of our findings are (1) an improved understanding of the relationship between nanoscale cellular organization and intracellular motion in stem cells may elucidate mechanisms and improve treatment strategies in regenerative medicine, and (2) the differentiation status of stem cells could potentially be determined and/or quantified in live cells in real time without the use of labels to assess stem cell population purity, spontaneous differentiation, and partial vs full differentiation.

To assess the potential of dual-PWS as a non-invasive real-time method to measure differentiation status, we have developed a machine learning model on this dataset to classify cells as hMSCs or osteoblasts based solely on physical properties of the nucleus. This model was developed using Scikit-learn Logistic Regression classifier⁹ and validated using a ten-fold cross validation scheme. This preliminary model has a classification ROC AUC of $88.6 \pm 4.4\%$ (SEM) and an accuracy of $80.4 \pm 3.4\%$ (SEM). It is likely that this model could achieve higher accuracy by increasing the size of the dataset and developing additional features based on the spatial distribution of these markers and analysis of temporal interference frequencies.”

6. The description of motion spectra of cells before and during fixation (lines 142-146) would be easier to interpret if referenced to a figure. I understand this was likely not referenced to figure 2 to keep the figure order correct, but in this case, this section should be moved to the corresponding results section.

We thank the reviewer for this suggestion, but we believe that it makes the most sense to keep this section where it was originally located. The goal of these lines and the two examples that follow are to compare hypothetical biological situations to theoretical changes in m_c , Φ , and the resulting effect on m_f . We believe that keeping all three examples together makes the most sense for the manuscript flow. To clarify this confusion, we have added a reference in this theoretical section to the experimental cellular fixation section and a reference in the experimental cellular fixation section to the theoretical description.

“However, the addition of chemically crosslinking paraformaldehyde would decrease the fractional moving mass by decreasing the volume fraction of moving chromatin without changing the density or length-scales (same m_c , small Φ , small m_f) (see **Cellular Fixation** section for experimental data).

...

The decrease in m_f is likely due to a decrease in Φ as hypothesized in the **Theory: Analysis of Temporal Interference** section.”

7. The statistical analysis is generally appropriate, but it would be helpful if p-values were indicated on the figures.

We thank the reviewer for this suggestion. P-values have been added to all of the figure.

Reviewer #3 (Remarks to the Author):

The authors present a multimodal label-free interferometric imaging platform for measuring intracellular nanoscale structure and macromolecular dynamics in living cells. Applying this system in vitro, they explore changes in higher-order chromatin structure and dynamics that occur due to cellular fixation, stem cell differentiation, and ultraviolet (UV) light irradiation. Finally, they discover a new phenomenon, cellular paroxysm, a near instantaneous, synchronous burst of motion that occurs early in the process of UV induced cell death.

During apoptosis, actin depolymerization is associated with caspase activation. That means that extensive degradation of molecules is also concomitant with cytoskeleton reorganization. In genuine apoptosis this process is accompanied by the formation of the apoptotic microtubule network and slight plasma membrane permeabilization.

Did the authors check if UV-induced apoptosis in this particular cell line is genuine apoptosis? Later when apoptotic microtubules depolymerized apoptotic cells undergo secondary necrosis and cell membrane become permeable for PI.

To me is not clear in which moment the authors detect the cellular paroxysm: during genuine apoptosis or when the cells undergo secondary necrosis.

The authors claim that “this burst of motion is connected to a disruption of the cytoskeletal and cell membrane structures” but during genuine apoptosis the plasma membrane is intact (PI negative, annexin V positive).

In Figure 7, plasma membrane is disrupted (PI positive) in UV irradiated cells. This process is accompanied by minimal incorporation of BrdU into the nuclei without cellular paroxysm. In these circumstances cells should be in secondary necrosis, however the nucleus is not fragmented suggesting that the cells after UV irradiation die by primary necrosis.

My impression is that UV-induced apoptosis needs better characterization.

Figure 6. What do MB and NMB stand for?

We thank the reviewer for this feedback. We understand and agree with the concern that it is not clear where the cellular paroxysm fits within the apoptotic/necrotic process. While it is clear that these cells are dying, the cells undergoing UV irradiation and cellular paroxysm don't seem to fit within the standard apoptotic/necrotic process. To avoid this confusion, we have added a discussion of UV irradiation and cellular paroxysm in the context of apoptosis/necrosis, a table comparing the features of apoptosis, necrosis, UV irradiation, and cellular paroxysm, as well as new figures, analysis, and experiments measuring mitochondrial membrane potential, caspase 3/7 activation, the effects of different UV dosages, cell recovery at longer timescales, and changes in cell morphology.

Additionally, we apologize for the confusion regarding MN and NMB in figure 6. This was a typo from a nomenclature in a previous draft; it has been corrected to CP (cellular paroxysm) and NCP (no cellular paroxysm).

Added to the main text:

“While these studies suggest that cellular paroxysm could be a never before seen early event in UV mediated cell death, the exact mechanism of cell death is unclear. UV irradiation is inducing cell death as observed through molecular processes such as PS externalization, membrane permeabilization, loss of mitochondrial membrane potential (**SI Figure 13**), and actin depolymerization, as well as a disruption in cell function as observed through a lack of intracellular motion [at the time of paroxysm and up to 20 hours later (*see SI UV Irradiation Length and Recovery; SI Figure 10*)], macroscopic cell movement within the field of view, and DNA replication (at time of paroxysm). The loss of mitochondrial membrane potential and the actin depolymerization are consistent with apoptotic and necrotic pathways, but some of these features do not follow the traditional pathways (**Table 1**). Typically, healthy viable cells are characterized by Annexin V-/PI-, cells in early apoptosis are characterized by Annexin V+/PI-, and necrotic or late stage apoptotic cells are characterized by Annexin V+/PI+. In these experiments, after UV irradiation, cells are exhibiting Annexin V-/PI+ before cellular paroxysm and Annexin V+/PI+ after cellular paroxysm. One possibility is that we are observing early necrotic cells in a short time window where the holes in the plasma membrane are large enough for PI to pass through, but not yet large enough for Annexin V. Otherwise, we are potentially observing a novel form of necrosis. Within traditional apoptosis and most pathways of necrosis, cells exhibit significant morphological changes, such as cell shrinkage, rounding, membrane blebbing, and eventual cellular fragmentation. While macroscopic morphological changes are observed in cells after short dosages of UV irradiation (1 min), no morphological changes are observed within 20 hours in cells irradiated for more than six minutes, with or without a cellular paroxysm (**SI Figure 15**). Additionally, activated caspase 3/7 was not detected in any of the irradiation conditions at 2.5 or 20 hours (**SI Figure 14**). In conclusion, while these cells are undergoing a form of cell death, they are not following traditional apoptotic or necrotic pathways (**Table 1**).”

Added to the SI:

“UV Irradiation and Markers of Cell Death

Reduction in mitochondrial membrane potential, activation of caspase 3/7, and alterations in cell morphology are common features in the process of cell death. These features were tested under various lengths of UV treatment (1 minute, 3 minutes, 6 minutes, and 20 minutes) and measured at ~3 and 20 hours after UV irradiation. We observed a dosage dependent decrease in mitochondrial membrane potential when measured 2.5 hours after irradiation (**SI Figure 13**).

Caspase 3/7 activation was not observed for any UV dosage at with 2.5 or 20 hours after UV irradiation (**SI Figure 14**). Note that while some signal is observed in these images, based on the localization and extremely low signal, the fluorescence seen in these images is likely to be autofluorescence and leakage from MitoTracker (Orange CMTMRos), not caspase 3/7 activation. Interestingly, morphological changes consistent with apoptosis were observed for low dosages of UV (1 and 3 minutes), but higher dosages of UV didn't show any change in morphology up to 20 hours later (**SI Figure 15**). None of these features show any correlation with the cellular paroxysm.”

Also added to the SI:

“UV Irradiation Length and Recovery

Continuous UV irradiation halts intracellular dynamics within 10-20 minutes, but it was unclear if the cell could recover from this damage and if that recovery would depend on the length of UV irradiation. To test this, HeLa cells were irradiated with UV for varying lengths of time (control, 30 seconds, 1 minute, 3 minutes, 6 minutes, and 20 minutes), and remeasured at 3 and 20 hours after UV irradiation. Cells that underwent 20 minutes of UV irradiation experienced cellular paroxysms, while the rest of the dosages did not. The response of cells undergoing 20 and 6 minutes of irradiation were similar, intracellular dynamics stopped, never recovered, and there were no changes in cellular morphology (**SI Figure 10**). The medium dosages of UV (1 and 3 minutes) also halted intracellular dynamics, but interestingly, some cells were able to restart their dynamics at 20 hours, while others were not (**SI Figure 11**). Additionally, these cells displayed morphological changes consistent with apoptosis: cell shrinkage, fragmentation, and detachment from the glass substrate. The 30 second dosage showed a slightly decrease in dynamics, but was able to completely recover (**SI Figure 12**). Overall, the length of UV irradiation affects the cellular response/recovery, and this recovery seems to be independent of the cellular paroxysm.”

Table 1

	PS Externalization	Membrane Permeability	Actin Depolymerization	Mitochondrial Membrane Potential	Caspase 3/7 Activation	Morphological Changes (< 3 hr)	Morphological Changes (12-24 hr)
>6 Min UV Irradiation + Cellular Paroxysm	Yes	Yes	High	Low*	No ^A	No	No
>6 Min UV Irradiation without Cellular Paroxysm	No	Yes	Medium	Low ^B	No ^A	No	No
1-3 Min UV Irradiation without Cellular Paroxysm	No	Not Tested	Not Tested	Medium*	No ^A	Yes	Yes
Early Stage Apoptosis	Yes	No	Medium	Low	Yes	No	NA
Last Stage Apoptosis	Yes	Yes	High	Low	No	NA	Yes
Necrosis	Yes	Yes	High	Low	No	Varied	Varied
Healthy Cells	No	No	Low	High	No	No	No

*Measured at 2.5 hours after UV irradiation

^Δ Measured at 2.5 and 20 hours after UV irradiation

“**Table 1.** Features of Cell Death and UV Induced Cellular Paroxysm. This table summarized different features of the apoptotic and necrotic processes and if these features occur in UV irradiated cells with and without cellular paroxysm.”

SI Figure 13

Mitochondrial Membrane Potential After UV Irradiation

“SI Figure 13: Mitochondrial Membrane Potential After UV Irradiation. MitoTracker Orange CMTMRos fluorescence images from HeLa cells 2.5 hours after various lengths of UV

irradiation (no irradiation, 1 minute, 3 minutes, and 6 minutes). UV irradiation reduces mitochondrial membrane potential.”

SI Figure 14

Caspase 3/7 after UV Irradiation

“SI Figure 14: Caspase 3/7 Activation After UV Irradiation. CellEvent Caspase-3/7 fluorescence images from HeLa cells 2.5 hours after various lengths of UV irradiation (no irradiation, 1 minute, 3 minutes, and 6 minutes). UV irradiated cells did not exhibit caspase 3/7 activation 2.5 hours after UV irradiation. While some signal is observed in these images, based on the localization and extremely low signal, the fluorescence seen in these images was determined to be autofluorescence and leakage from MitoTracker, not caspase 3/7 activation.”

SI Figure 15

Morphological Alterations After UV Irradiation

“SI Figure 15: Morphological Changes Due To UV Irradiation. Representative reflectance microscopy images showing morphological changes from HeLa cells that were irradiated with UV for 1 minute (top), 6 minutes (middle), and 20 minutes (bottom) at time points before

irradiation (left), 3 hours after irradiation (middle), and 20 hours after irradiation (right). Low dosage UV irradiation induces morphological changes consistent with apoptosis, while cells undergoing higher UV dosages don't show any morphological changes up to 20 hours later.”

SI Figure 10

“SI Figure 10: Long Dosage UV Irradiation and Recovery. Representative m_f maps of HeLa cells that were irradiated with UV for 6 minutes (top) and 20 minutes (bottom) at time points before irradiation (left), 3 hours after irradiation (middle), and 20 hours after irradiation (right). After long dosages of UV cells are unable to recover any macromolecular motion up to 20 hours later.”

SI Figure 11

“SI Figure 11: Medium Dosage UV Irradiation and Recovery. Representative m_f maps of HeLa cells that were irradiated with UV for 1 minute (top) and 3 minutes (bottom) at time points before irradiation (left), 3 hours after irradiation (middle), and 20 hours after irradiation (right). After medium dosages of UV some cells are able to partially recover macromolecular motion.”

SI Figure 12

“SI Figure 12: Control and Low Dosage UV Irradiation and Recovery. Representative m_f maps of HeLa cells didn’t undergo irradiation (top) and cells that were irradiated with UV for 30 seconds (bottom) at time points before irradiation (left), 3 hours after irradiation (middle), and 20 hours after irradiation (right). Control cells and low dosage UV irradiation do not cause significant changes in macromolecular motion.”

Rebuttal References

- 1 Park, Y., Depeursinge, C. & Popescu, G. Quantitative phase imaging in biomedicine. *Nature Photonics* **12**, 578-589, doi:10.1038/s41566-018-0253-x (2018).
- 2 Ma, L. *et al.* Phase correlation imaging of unlabeled cell dynamics. *Scientific Reports* **6**, 32702, doi:10.1038/srep32702
<https://www.nature.com/articles/srep32702#supplementary-information> (2016).
- 3 Majeed, H. *et al.* Magnified Image Spatial Spectrum (MISS) microscopy for nanometer and millisecond scale label-free imaging. *Opt. Express* **26**, 5423-5440, doi:10.1364/OE.26.005423 (2018).

- 4 Wang, R. *et al.* Dispersion-relation phase spectroscopy of intracellular transport. *Opt. Express* **19**, 20571-20579, doi:10.1364/OE.19.020571 (2011).
- 5 Chandler, J. E., Cherkezyan, L., Subramanian, H. & Backman, V. Nanoscale refractive index fluctuations detected via sparse spectral microscopy. *Biomedical optics express* **7**, 883-893, doi:10.1364/boe.7.000883 (2016).
- 6 Hu, C. & Popescu, G. Physical significance of backscattering phase measurements. *Opt. Lett.* **42**, 4643-4646, doi:10.1364/OL.42.004643 (2017).
- 7 Kaur, G. *et al.* Probing transcription factor diffusion dynamics in the living mammalian embryo with photoactivatable fluorescence correlation spectroscopy. *Nature Communications* **4**, 1637, doi:10.1038/ncomms2657
<https://www.nature.com/articles/ncomms2657#supplementary-information> (2013).
- 8 Dixon, J. R. *et al.* Chromatin architecture reorganization during stem cell differentiation. *Nature* **518**, 331, doi:10.1038/nature14222
<https://www.nature.com/articles/nature14222#supplementary-information> (2015).
- 9 Pedregosa, F. a. V., G. and Gramfort, A. and Michel, V. and Thirion, B. and Grisel, O. and Blondel, M. and Prettenhofer, P. and Weiss, R. and Dubourg, V. and Vanderplas, J. and Passos, A. and Cournapeau, D. and Brucher, M. and Perrot, M. and Duchesnay, E. Scikit-learn: Machine Learning in Python. *Journal of Machine Learning Research* **12**, 2825-2830 (2011).

REVIEWERS' COMMENTS:

Reviewer #1 (Remarks to the Author):

The authors addressed my comments in detail. A couple of points left:

1. The optical setup, new Fig. 1: I doubt that the system really looks like that, because the LED source seems to flood the detector. I suspect the LED is directly opposite from the sample (at the bottom of the figure) and the beam splitter (rotated 90deg to the right) then sends some backscattered light to the CCD. Also, in the text the authors refer to a "lamp" instead of LED. There is also mention of a condensor other than the objective used to focus light onto the sample that seems inconsistent with the setup figure?
2. The authors should clarify that this system is not interferometric and that the calculations assume small angle approximations. Equation 16 assumes zero numerical aperture, meaning, a momentum transfer of $2k$. This is the DLS result for far-field backscattering measurements. This of course means, loss of any transverse resolution.
3. What is the benefit of using a spectrometer at all? The diffusion coefficient can be obtained directly at a single wavelength.

Reviewer #2 (Remarks to the Author):

The revised manuscript has generally responded adequately to my original concerns. I have two followup comments that should be addressed prior to publication:

1. DLS microscopy, including images of decay rate and mobile fractions has been performed previously on living cells (see ref 1 below). The differences and drawbacks relative to the present method should be described.
2. Multiple figures (both original and newly added) are missing quantitative scale bars and labels. Addition of these would help add context to the time and length scales of the observed motions and enable better comparison to previous work. Ex. SI Figure 7 shows dynamic motion, but velocity cannot be estimated without length and time scales.
 - a. Time labels should be added to either figures or the accompanying captions anyplace where successive images of the same live cell are shown. As one example, the new SI figures (ex. SI Figure 7-9) accompany discussion of the temporal dynamics of cellular motion, but this is difficult to gauge without labeled times between frames.
 - b. Length scale bars are not present on all images (or at least one per set at the same scale).
 - c. Multiple figures throughout the manuscript use pseudocolor to indicate quantitative data, but there are very few quantitative scale bars. This would help in the interpretation of the data and comparison to previous work (ex. ref 1 below).

references

1. Dzakpasu, R. and Axelrod, D., "Dynamic light scattering microscopy. A novel optical technique to image submicroscopic motions. II: Experimental applications." *Biophysical Journal* 87, 1288-1297 (2004).

Reviewer #3 (Remarks to the Author):

The authors have addressed sufficiently the main concerns of my previous review.

We want to thank all the reviewers for their helpful feedback and suggestions.

REVIEWERS' COMMENTS:

Reviewer #1 (Remarks to the Author):

The authors addressed my comments in detail. A couple of points left:

1. The optical setup, new Fig. 1: I doubt that the system really looks like that, because the LED source seems to flood the detector. I suspect the LED is directly opposite from the sample (at the bottom of the figure) and the beam splitter (rotated 90deg to the right) then sends some backscattered light to the CCD. Also, in the text the authors refer to a "lamp" instead of LED. There is also mention of a condenser other than the objective used to focus light onto the sample- that seems inconsistent with the setup figure?

We thank the reviewer for their helpful feedback and suggestions. We have modified Figure 1 based on their suggestions to more accurately reflect our system design. All references to a “lamp” have been replaced with LED. Finally, we are confused by this mention of a condenser; it is unclear where in the text a condenser is mentioned. We have modified the text to clarify that the light is being focused by the objective onto the sample.

From the main text:

“With the microscope in epi-illumination mode, light from the LED is passed through a long pass filter to remove UV components before being focused by the objective onto the sample with an illumination NA of 0.55. The backscattered light is collected by the objective, passed through the spectral filter (LCTF) and imaged at the CCD (Figure 1a).”

“To perform UV irradiation, the cells were illuminated with the full LED spectra...”

“Temporal interference data cube is loaded and normalized for LED intensity, exposure time, and dark counts.”

“To correct for temporal LED fluctuations (and other system noise)...”

From the Supplementary Information:

“*LED Intensity*: SNR can be directly increased by increasing the input LED intensity.”

Figure 1

a) Dual-PWS Instrumentation

b) Dual-PWS Acquisition and Analysis

“Figure 1: Dual-PWS Instrumentation, Acquisition, and Analysis. (a) Schematic of the Dual-PWS instrumentation. Broadband white light from a light-emitting diode (LED) is passed through a filter to remove ultraviolet (UV) components before being focused onto the sample. The backscattered light is collect, spectrally filtered through a liquid crystal tunable filter (LCTF), and imaged with a charge-coupled device (CCD) camera. (b) (left) Structural measurements (Σ_s) are collected by acquiring multiple backscattered wide-field monochromatic images across a range of wavelengths (500-700nm) to produce a three-dimensional image cube, $I(\lambda, x, y)$, where λ is the wavelength and (x, y) correspond to pixel. Σ_s is extracted by calculating the standard deviation of the spectral interference at each pixel (x, y) . (right) Dynamics measurements [Σ_t^2 , m_f (fractional moving mass), and D (diffusion coefficient)] are collected by acquiring multiple backscattered wide-field images at a single wavelength (550nm) over a period of time (acquisition time), to produce a three-dimensional image cube, $I(t, x, y)$, where t is time and (x, y) correspond to pixel positions. D is extracted by calculating the decay rate of the autocorrelation of the temporal interference and m_f is calculated by normalizing the variance of the temporal interference (Σ_t^2) at each pixel (x, y) .”

2. The authors should clarify that this system is not interferometric and that the calculations assume small angle approximations. Equation 16 assumes zero numerical aperture, meaning, a momentum transfer of $2k$. This is the DLS result for far-field backscattering measurements. This of course means, loss of any transverse resolution.

We thank the reviewer for these suggestions. The dynamics and structural measurements acquired by dual-PWS are extracted from signals originating from the physics of light interference. We understand that some readers may associate the term “interferometric” to specifically relate to techniques based on thin film interference, such as quantitative phase imaging or optical profilometry. To avoid this confusion, we will change all references in both the text and manuscript title from “interferometric” to “interference-based”.

New Manuscript Title:

“Multimodal interference-based imaging of nanoscale structure and macromolecular motion uncovers UV induced cellular paroxysm”

Changes to the abstract:

“Here we present a multimodal label-free imaging platform for measuring intracellular structure and macromolecular dynamics in living cells with a sensitivity to macromolecules as small as 20nm and millisecond temporal resolution.”

Changes to the main text:

“Building upon these advancements, we present a label-free interference-based platform (dual-PWS) that captures the temporal behavior and structural organization of macromolecular assemblies in live cells.”

“Consequently, this interference-based platform measures nanoscale macromolecular motion in tandem with spatio-temporal behavior of the macromolecular ultrastructure in dozens of live cells simultaneously without photobleaching artifacts.”

Additionally, the theory does assume a small, but non-zero NA (small angle approximation). We have clarified this assumption in Supplementary Note 1 (Temporal Interference Theory) section. Note that $B(x', y')$ does have transverse resolution. In the theory, when calculating the ensemble average, this transverse resolution is somewhat lost, but in practice we do have transverse resolution as our system is producing images.

Changes to Supplementary Note 1 (Temporal Interference Theory):

”When the object is imaged by an epi-illumination bright-field microscope with a small NA (small angle approximation)...”

3. What is the benefit of using a spectrometer at all? The diffusion coefficient can be obtained directly at a single wavelength.

This is a good question. The dynamics measurements reported in this manuscript could be acquired with a system that replaces the spectral filter (LCTF) with a narrow bandpass filter. Although, one of the unique advantages of this system is that it can acquire both measurements of nanoscale structural organization and macromolecular dynamics within seconds. A spectral filter or spectrometer is required to acquire the structural measurements of the dual-PWS system. Additionally, since both of these measurements are captured through the same filter/camera/lightpath, they can be perfectly co-localized to accurately compare structure and dynamics at the same location. Finally, while this is largely not explored in the current manuscript, the correlation function does depend on the wavelength of light; this could be exploited by acquiring dynamics measurements at multiple different wavelengths, enabling extraction of additional dynamics information.

Reviewer #2 (Remarks to the Author):

The revised manuscript has generally responded adequately to my original concerns. I have two followup comments that should be addressed prior to publication:

1. DLS microscopy, including images of decay rate and mobile fractions has been performed previously on living cells (see ref 1 below). The differences and drawbacks relative to the present method should be described.

Thank you for the question. There are several similarities between dual-mode PWS and DLS microscopy. Largely, they are both label-free imaging techniques that measure macromolecular dynamics in live cells. While both techniques measure diffusion coefficients, there are differences between the mobile fraction measured by DLS microscopy and fractional moving mass measured by dual-mode PWS. From Dzakpasu et al, the mobile fraction is the fraction of scattering centers that are mobile (vs immobile). Fractional moving mass (in units of mass) provides a measurement of the volume fraction of moving structures and the mass of typically moving structures. Volume fraction here is the ratio of mobile structures over imaging volume compared to mobile fraction, which is the ratio of mobile scattering structures over immobile scattering structures. Experimentally, dual-PWS has shown sensitivity to nanospheres an order of magnitude smaller than DLS microscopy as reported by Dzakpasu et al (25nm vs 200nm). One of the biggest differences is that dual-mode PWS integrates and co-localized measurements of intracellular dynamics with macromolecular structure providing a much more thorough understanding of the physical state of the cell. The integration of this information has enabled new applications, such as the identification of stem cells presented in this manuscript, which would not have been nearly as informative (or accurate) with macromolecular dynamics measurements alone.

We have added reference to DLS microscopy to our introduction of label free dynamics techniques:

“To address these issues, techniques have been developed based on quantitative phase imaging (QPI)¹⁰ and dynamic light scattering (DLS)¹¹ to image intracellular dynamics without the use of labels.”

Many of the other comparisons are already addressed in the manuscript, but we have modified the text to clarify and elaborate on these differences.

“By combining these two techniques, we pair measurements of cellular dynamics with macromolecular structure – creating a dual light interference platform (dual-PWS) to greatly enhance our understanding of the physical state of the cell and our ability to probe cellular behavior at the level of macromolecular assemblies.”

“In addition to measuring diffusion coefficients (similar to other label-free techniques such as QPI and DLS), dual-PWS extracts additional quantifications of motion such as the fractional moving mass, which provides a measurement of the volume fraction of moving structures and the mass of moving structures.”

“Integrating measurements of nanoscale structure and macromolecular motion has provided an enhanced understand of the physical state of the cell enabling applications such as classification of stem cell differentiation status.”

2. Multiple figures (both original and newly added) are missing quantitative scale bars and labels. Addition of these would help add context to the time and length scales of the observed motions and enable better comparison to previous work. Ex. SI Figure 7 shows dynamic motion, but velocity cannot be estimated without length and time scales.

a. Time labels should be added to either figures or the accompanying captions anyplace where

successive images of the same live cell are shown. As one example, the new SI figures (ex. SI Figure 7-9) accompany discussion of the temporal dynamics of cellular motion, but this is difficult to gauge without labeled times between frames.

b. Length scale bars are not present on all images (or at least one per set at the same scale).

c. Multiple figures throughout the manuscript use pseudocolor to indicate quantitative data, but there are very few quantitative scale bars. This would help in the interpretation of the data and comparison to previous work (ex. ref 1 below).

Thank you for these suggestions. We have added time labels to SI Figures 7, 8, and 9. Length scale bars have been added to Figures 6, 7, 8 and SI Figures 6, 7, 8, 9, 10, 11, 12, 13, 14, and 15. Quantitative pseudocolor scale bars have been added to Figures 3, 4, 6 and SI Figures 6, 7, 8, 9, 13, 14, and 15. Note that we did not add a quantitative pseudocolor scale bar to the Sigma maps (top row) of figure 7. We believe that it would not improve the message of the figure and would negatively impact the aesthetic symmetry. As a compromise, we have added the quantitative scale bounds to the figure 7 legend text.

Figure 3

Figure 3: Cellular Fixation. a) Representative m_f (top) and Σ_s (bottom) maps of HeLa cells before and after crosslinking fixation using paraformaldehyde (PFA). b) Bar graphs with individual data points quantifying the mean m_f and Σ_s for the nucleus and cytoplasm separately. Scale bar is 20 μm . Reported p-values are from two-tailed, paired Student's t-tests.

Figure 4

Figure 4: Stem Cell Differentiation. a) Representative Σ_s (top), m_f (middle), and D (bottom) maps of human mesenchymal stem cells (left) and osteoblasts (right) with colored region used to show nuclear segmentation. b) Box plots showing the distribution of Σ_s , m_f , and D for the stem cells and osteoblasts. We observe different states of chromatin folding and nuclear dynamics comparing hMSCs to osteoblasts. Scale bar is 8 μm . The box within the plot represents the 1st quartile, median (center line), and 3rd quartile; the whiskers extend to the minimum and maximum values excluding any outliers (1.5x the interquartile range above the 3rd quartile or below the 1st quartile; outliers not shown). Reported p-values are from two-tailed, heteroscedastic Student's t-tests.

Figure 6

Figure 6: Cellular Paroxysm Temporal Analysis. (a) Representative m_f maps of a cell undergoing the cellular paroxysm. (b) Example temporal spectra from a single pixel during a cellular paroxysm. Using spectral feature detection the timing of the event can be determined for each pixel reducing the temporal resolution from the full acquisition time ($\sim 7s$) to a single exposure time (35ms). (c) Example histogram of the timing of the event for each pixel within a single cell. (d) Map displaying the timing of the event for each pixel within a representative cell. The spatial distribution of this timing shows that the initial motion occurs simultaneously within a single 35ms frame on opposite ends of the cell. Scale bar is $7 \mu m$.

Figure 7

Figure 7: Cellular Paroxysm Biological Exploration. Exploration of cytoskeletal (a) and membrane (b) disruption during cellular paroxysm using Alexa Fluor 488 Phalloidin and Annexin V-FITC respectively. (row 1) Representative Σ_s maps of cells before UV irradiation, scaled from 0.00012 – 0.00079. (row 2) Multiple m_f maps processed and combined to show which cells experienced the cellular paroxysm event throughout the experiment. (row 3) Representative fluorescence image of Fluor 488 Phalloidin (a) and Annexin V-FITC (b). (row 4) (a) Box plot showing distribution of the normalized phalloidin intensity and (b) bar graph counting cells that are grouped based on their Annexin V state [positive (Ann+) or negative (Ann-)] and whether they experienced the cellular paroxysm (CP) or not (NCP). The cellular paroxysm is correlated to both cytoskeletal and membrane disruption. The box within the plot represents the 1st quartile, median (center line), and 3rd quartile; the whiskers extend to the minimum and maximum values excluding any outliers (1.5x the interquartile range above the 3rd quartile or below the 1st quartile; outliers not shown). Scale bar is 11 μ m. Reported p-values are from two-tailed, heteroscedastic Student's t-tests.

Figure 8

Figure 8: Membrane Poration and DNA Replication. (a) Representative fluorescence images of propidium iodide (PI) for control cells (top) and UV irradiated cells (bottom) show that pores form in the cell membrane in response to UV irradiation. (b) Representative fluorescence images of Bromodeoxyuridine (BrdU) for control cells (top) and UV irradiated cells (bottom) show that UV irradiation is stalling DNA replication. Scale bar is 5 μm .

Supplementary Figure 6

Partial Cellular Paroxysms

Supplementary Figure 6: Partial Cellular Paroxysms. Partial paroxysms are one of the features identified during the characterization of cellular paroxysms. Generally, paroxysms occur within the entirety of the cell, but occasionally, they will only effect a small section of the cell. Scale bar is 8 μm .

Supplementary Figure 7

Wavelike Initiation of Cellular Paroxysms

Supplementary Figure 7: Wavelike Initiation of Cellular Paroxysms. Wavelike initiation of the cellular paroxysm is one of the features identified during the characterization of cellular paroxysms. Generally, paroxysms initiate across the entirety of the cell instantaneously, but occasionally, they will initiate in one part of the cell and spread out in a wavelike manner across the entire cell. Scale bar is 8 μm .

Supplementary Figure 8

Synchronous Cellular Paroxysms of Adjacent Cells

Supplementary Figure 8: Synchronous Cellular Paroxysms. Synchronous cellular paroxysms are one of the features identified during the characterization of cellular paroxysm. Generally,

paroxysms initiate asynchronously from cell to cell, but occasionally, adjacent cells (generally in contact with each other) will undergo cellular paroxysm simultaneously. Scale bar is 8 μm .

Supplementary Figure 9

Cyclical Cellular Paroxysms

Supplementary Figure 9: Cyclical Cellular Paroxysms. Cyclical cellular paroxysms are one of the features identified during the characterization of cellular paroxysm. Sometimes, after the initial cellular paroxysm, a secondary smaller paroxysm will occur milliseconds to tens of seconds after the initial paroxysm has subdued. Scale bar is 8 μm .

Supplementary Figure 10

Mitochondrial Membrane Potential After UV Irradiation

Supplementary Figure 10: Mitochondrial Membrane Potential After UV Irradiation. MitoTracker Orange CMTMRos fluorescence images from HeLa cells 2.5 hours after various lengths of UV irradiation (no irradiation, 1 minute, 3 minutes, and 6 minutes). UV irradiation reduces mitochondrial membrane potential. Scale bar is 13 μm .

Supplementary Figure 11

Caspase 3/7 after UV Irradiation

Supplementary Figure 11: Caspase 3/7 Activation After UV Irradiation. CellEvent Caspase-3/7 fluorescence images from HeLa cells 2.5 hours after various lengths of UV irradiation (no irradiation, 1 minute, 3 minutes, and 6 minutes). UV irradiated cells did not exhibit caspase 3/7 activation 2.5 hours after UV irradiation. While some signal is observed in these images, based on the localization and extremely low signal, the fluorescence seen in these images is likely to be autofluorescence and leakage from MitoTracker, not caspase 3/7 activation. Scale bar is 13 μm .

Morphological Alterations After UV Irradiation

Initial

3 Hours

20 Hours

Supplementary Figure 12: Morphological Changes Due To UV Irradiation. Representative reflectance microscopy images showing morphological changes from HeLa cells that were irradiated with UV for 1 minute (top), 6 minutes (middle), and 20 minutes (bottom) at time points before irradiation (left), 3 hours after irradiation (middle), and 20 hours after irradiation (right). Low dosage UV irradiation induces morphological changes consistent with apoptosis, while cells undergoing higher UV dosages don't show any morphological changes up to 20 hours later. Scale bar is 19 μm .

Supplementary Figure 13

Supplementary Figure 13: Long Dosage UV Irradiation and Recovery. Representative m_f maps of HeLa cells that were irradiated with UV for 6 minutes (top) and 20 minutes (bottom) at time points before irradiation (left), 3 hours after irradiation (middle), and 20 hours after irradiation (right). After long dosages of UV cells are unable to recover any macromolecular motion up to 20 hours later. Scale bar is 19 μm .

Supplementary Figure 14

Supplementary Figure 14: Medium Dosage UV Irradiation and Recovery. Representative m_f maps of HeLa cells that were irradiated with UV for 1 minute (top) and 3 minutes (bottom) at time points before irradiation (left), 3 hours after irradiation (middle), and 20 hours after irradiation (right). After medium dosages of UV some cells are able to partially recover macromolecular motion. Scale bar is 19 μm .

Supplementary Figure 15

Supplementary Figure 15: Control and Low Dosage UV Irradiation and Recovery. Representative m_f maps of HeLa cells didn't undergo irradiation (top) and cells that were irradiated with UV for 30 seconds (bottom) at time points before irradiation (left), 3 hours after irradiation (middle), and 20 hours after irradiation (right). Control cells and low dosage UV irradiation do not cause significant changes in macromolecular motion. Scale bar is 19 μm .

references

1. Dzakpasu, R. and Axelrod, D., "Dynamic light scattering microscopy. A novel optical technique to image submicroscopic motions. II: Experimental applications." *Biophysical Journal* 87, 1288-1297 (2004).

Reviewer #3 (Remarks to the Author):

The authors have addressed sufficiently the main concerns of my previous review.